# REASONING-DRIVEN MULTIMODAL LLMS FOR DOMAIN GENERALIZATION

**Zhipeng Xu**[1,*]**, Zilong Wang**[2,*]**, Xinyang Jiang**[2,*]**, Dongsheng Li**[2]**, De Cheng**[1,3,✉]**, Nannan Wang**[1]

[1]Xidian University,  [2]Microsoft Research Asia
[3]National Engineering Laboratory for Integrated Aero-Space-Ground- Ocean Big Data Application
`xu_zhipeng@stu.xidian.edu.cn`, `{nnwang,dcheng}@xidian.edu.cn`
`{wangzilong, xinyangjiang, dongsli}@microsoft.com`

## ABSTRACT

This paper addresses the domain generalization (DG) problem in deep learning. While most DG methods focus on enforcing visual feature invariance, we leverage the reasoning capability of multimodal large language models (MLLMs) and explore the potential of constructing reasoning chains that derives image categories to achieve more robust predictions under domain shift. To this end, we systematically study the role of reasoning in DG using DomainBed-Reasoning, a newly constructed extension of DomainBed dataset, in which each sample is paired with class-relevant reasoning chains. Our analysis reveals two key challenges: (i) fine-tuning MLLMs with reasoning chains for classification is more challenging than direct label supervision, since the model must optimize complex reasoning sequences before label prediction; and (ii) mismatches in reasoning patterns between supervision signals and fine-tuned MLLMs lead to a trade-off between semantic richness (informative but harder to optimize) and optimization efficiency (easier to optimize but less informative). To address these issues, we propose RD-MLDG (Reasoning-Driven Multimodal LLM for Domain Generalization), a framework with two components: (i) MTCT (Multi-Task Cross-Training), which introduces an additional direct classification pathway to guide reasoning supervision; and (ii) SARR (Self-Aligned Reasoning Regularization), which preserves the semantic richness of reasoning chains while mitigating reasoning-pattern mismatches via iterative self-labeling. Experiments on standard DomainBed datasets (PACS, VLCS, OfficeHome, TerraInc) demonstrate that RD-MLDG achieves state-of-the-art performances, highlighting reasoning as a promising complementary signal for robust out-of-domain generalization.

## 1 INTRODUCTION

In real-world scenarios, deep learning models are often required to generalize reliably to unseen distributions. However, due to domain shift, their performance often degrades substantially when deployed in new environments. To address this issue, Domain Generalization (DG) has emerged as a key research area, aiming to learn representations and prediction functions that transfer well to unseen domains using only source-domain data.

Existing DG approaches generally fall into four categories: invariant representation learning, data augmentation, regularization, and meta-learning. While effective, these methods primarily focus on feature-level invariance and often fail to capture higher-level cross-domain commonalities, limiting their ability to generalize in complex scenarios.

Recently, multi-modal large language models (MLLMs) have made significant progress and demonstrate strong reasoning capabilities. This capability offers a new opportunity for DG: rather than

---

[1]✉ Corresponding author: De Cheng (dcheng@xidian.edu.cn)
[2]∗ Equal contribution: Zhipeng Xu, Zilong Wang and Xinyang Jiang contributed equally to this work.
[3]Work done during an internship at Microsoft Research Asia.

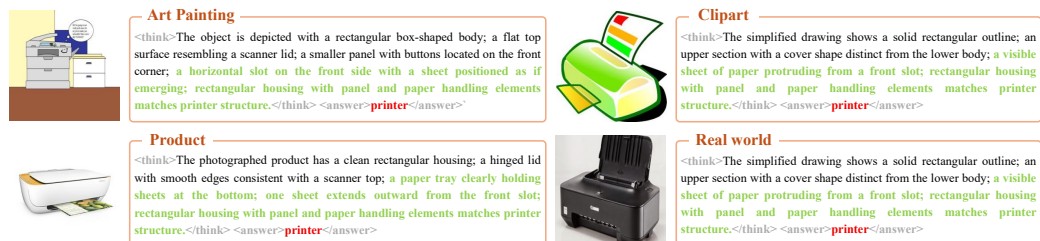

Figure 1: Illustrative examples of the printer in OfficeHome (Art Painting, Clipart, Product, Real World). Although the visual appearance differs substantially, the highlighted green segments of the reasoning chains remain highly consistent, capturing class-relevant cues that are likely to generalize well across domains.

relying solely on invariant feature representations, one could exploit reasoning chains to bridge domain gaps and achieve more robust generalization. By producing structured, class-relevant reasoning chains, we can leverage MLLMs to explicitly decompose a classification task into interpretable and domain invariant steps with strong transferability across domains (Fig. 1). To systematically study the role of reasoning in DG, we leverage GPT-4o to construct DomainBed-Reasoning, an extension of the widely used DomainBed benchmark, where each sample is paired with class-relevant reasoning chains. Specifically, we investigate whether an MLLM fine-tuned with DomainBed-Reasoning can leverage generated reasoning chains to improve domain-generalized predictions.

However, our empirical study shows that simply applying reasoning-chain supervision to classification tasks remains challenging. Our analysis reveals two key issues. First, reasoning-chain supervision is inherently difficult to optimize. In contrast to direct label prediction, the model must first fit complex reasoning chains before producing the final label. Second, reasoning chains generated by commercial models and those produced by fine-tuned MLLMs often exhibit distinct reasoning patterns. Commercial-model chains usually contain detailed contextual information that can assist classification, such as descriptions of background, object attributes, or scene conditions. While these chains are semantically rich and potentially informative, they are difficult for the model to optimize. In contrast, self-generated chains from the fine-tuned MLLM tend to be simpler in logic and more label-focused, which makes them easier to fit but less informative for classification. These observations indicate that straightforward reasoning supervision is insufficient and instead call for a more principled integration strategy.

This motivates us to design RD-MLDG (Reasoning-Driven Multimodal LLM for Domain Generalization), the first DG framework that explicitly incorporates class-relevant reasoning chains to enhance out-of-domain performance. RD-MLDG consists of two key components: (i) MTCT (Multi-Task Cross-Training), which jointly optimizes a direct classification pathway and a reasoning-augmented pathway. The direct pathway focuses on standard label prediction, offering a simple and stable training signal. This signal guides the reasoning pathway by preventing it from fitting to overly complex or inconsistent reasoning chains, and instead keeps the supervision focused on the core classification objective; and (ii) Self-Aligned Reasoning Regularization (SARR), which retains informative reasoning chains while mitigating reasoning-pattern mismatches through self-generated supervision. Together, these components enable RD-MLDG to provide a principled way to integrate reasoning into DG for robust generalization.

In summary, our work makes the following contributions:

- We construct DomainBed-Reasoning, an extension of the DomainBed dataset where each sample is paired with class-relevant reasoning chains, enabling systematic evaluation of both classification accuracy and reasoning consistency under domain shift.
- We propose RD-MLDG, the first DG framework that explicitly integrates reasoning by addressing two key challenges: (i) the difficulty of optimizing reasoning-chain supervision, and (ii) mismatches in reasoning patterns between supervision signals and fine-tuned MLLMs.
- We empirically show that RD-MLDG achieves state-of-the-art performance on standard DomainBed benchmarks (PACS, VLCS, OfficeHome, TerraInc), demonstrating the effectiveness of reasoning-based approaches for DG.

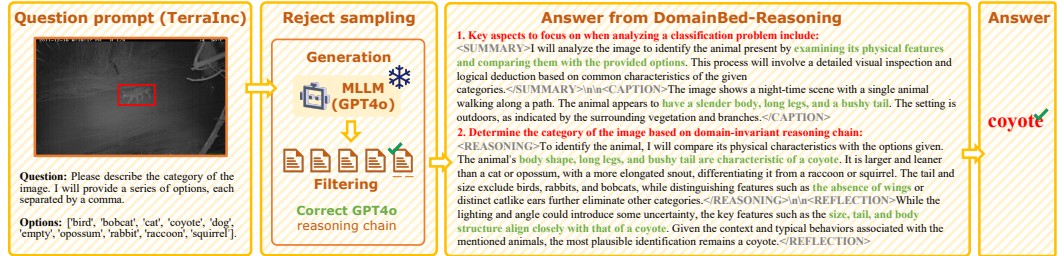

Figure 2: Overview of the DomainBed-Reasoning construction pipeline. GPT-4o generates multi-stage reasoning chains (`<SUMMARY>`, `<CAPTION>`, `<REASONING>`, `<REFLECTION>`, `<CONCLUSION>`) without access to ground-truth labels. Multiple candidates are sampled and filtered through rejection sampling to obtain coherent reasoning chains, which form the foundation for analyzing reasoning challenges in DG.

## 2 RELATED WORK

**Domain generalization (DG)** has been extensively studied through approaches such as invariant representation learning (e.g., IRM (Arjovsky et al., 2019), CORAL (Sun & Saenko, 2016)), data augmentation (e.g., MixStyle (Zhou et al., 2021), RandConv (Choi et al., 2023)), and learning-based strategies including meta-learning (e.g., MLDG (Li et al., 2018a)), adversarial training (e.g., PAPT (Xu et al., 2025)), and flat-minima regularization (e.g., SWAD (Cha et al., 2021)). Despite their success, these methods primarily operate at the representation level. More recently, CLIP-powered models have advanced DG by providing rich multimodal representations and strong cross-domain transfer. Recent efforts mainly focus on (i) prompt optimization (e.g., CoOp (Zhou et al., 2022b), CoCoOp (Zhou et al., 2022a), KgCoOp (Yao et al., 2023), MaPLe (Khattak et al., 2023), DPR (Cheng et al., 2024), PADG (Cheng et al., 2026), SECA (He et al., 2025b)) and (ii) leveraging CLIP as a backbone for robustness (e.g., StyLIP (Bose et al., 2024), PromptStyler (Cho et al., 2023)). Other works (He et al., 2025a; Cheng et al., 2025; Wang et al., 2025b;a) have also made meaningful progress in this direction. VOLDOGER (Choi et al., 2025) focuses on DG data annotation for vision-language tasks, such as image captioning, visual question answering (VQA), and visual entailment (VE). It alleviates the burden of manual annotation by extending LLM-based annotation techniques and provides a reliable benchmark for evaluating DG in these tasks. While effective, these methods remain confined to feature-level invariance and overlook the reasoning processes that underlie generalizable decision-making. Our work addresses this gap by introducing reasoning as a complementary signal for DG. Specifically, we aim to encourage process-level invariance through class-relevant reasoning chains, which serve as complementary signals to feature-level invariance for improving out-of-domain generalization.

**Multimodal large language models (MLLMs)** have recently emerged as a dominant paradigm for vision–language understanding, achieving strong reasoning (Yang et al., 2025) and generalization across modalities (e.g., Flamingo (Alayrac et al., 2022), GPT-4V (OpenAI, 2023), Gemini (Team et al., 2023)). Despite these advances, recent studies show that MLLMs underperform their vision encoders (e.g., CLIP (Radford et al., 2021b), EVA-CLIP (Sun et al., 2023)) on fundamental classification benchmarks such as ImageNet, Flowers102, and StanfordCars (Zhang et al., 2024). Existing improvements remain rooted in label-level supervision and fail to capture the explicit reasoning processes required for robust generalization under distribution shift. Our work addresses this gap by leveraging class-relevant reasoning chains and introducing effective training strategies to make reasoning supervision both optimizable and informative, thereby bridging the divide between feature-level alignment and process-level reasoning in MLLMs.

## 3 DOMAINBED-REASONING

A central challenge in studying reasoning for domain generalization (DG) is the lack of benchmarks that explicitly capture reasoning processes. Existing DG datasets provide only input–label pairs, which limits evaluation to feature-level invariance and makes it difficult to assess how reasoning contributes to generalization under domain shift.

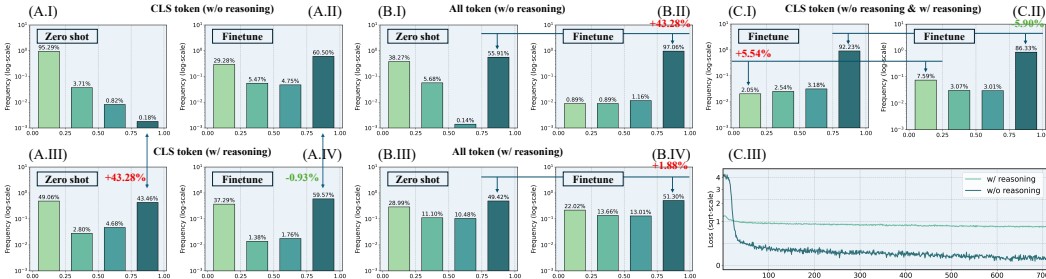

Figure 3: Illustration of Challenge 1 (see Sec. 4.1). (A) Classification token probabilities on target-domain test data under zero-shot and SFT, with and without reasoning; (B) Probability distributions of all tokens on source-domain training data under zero-shot and SFT; (C) Training dynamics: (I–II) classification token probabilities on source-domain training data after SFT; (III) loss curves comparing reasoning-based and no-thinking SFT.

To address this gap, we construct **DomainBed-Reasoning**, an extension of the widely used DomainBed benchmark enriched with structured reasoning chains. Specifically, for each image–label pair across four standard DG datasets (PACS, VLCS, OfficeHome, and TerraInc), we employ GPT-4o to generate multi-stage reasoning chains consisting of a `<SUMMARY>`, `<CAPTION>`, `<REASONING>`, `<REFLECTION>`, and `<CONCLUSION>`. Compared to the four-step format in LLaVA-CoT (Xu et al., 2024), we introduce an additional reflection stage that prompts the model to self-check intermediate reasoning, which empirically reduces invalid generations and improves stability. Importantly, the ground-truth label is withheld during generation, encouraging reasoning that is grounded in visual evidence rather than rote restatement.

Generating reasoning chains without access to the ground-truth label is non-trivial and may result in incomplete or inconsistent outputs. To improve quality, we adopt a rejection sampling strategy: for each instance, multiple candidate chains are generated, and only those that contain all required components and produce a coherent conclusion are retained. An overview of this construction pipeline is illustrated in Fig. 2. The resulting dataset pairs classification labels with independently generated reasoning chains, enabling evaluation under domain shift. DomainBed-Reasoning thus provides a standardized extension of DomainBed for studying reasoning-based domain generalization, serving as a foundation to analyze how reasoning supervision affects out-of-domain performance (Sec. 4).

# 4 CHALLENGES OF REASONING CHAIN IN DG

DomainBed-Reasoning provides a basis to analyze the role of reasoning in DG, and our study indicates that using reasoning-chain supervision alone for classification is challenging, especially under supervised fine-tuning. We identify two main issues: (i) the difficulty of optimizing reasoning-chain supervision under domain shift (Sec. 4.1), and (ii) mismatches in reasoning patterns between supervision signals and fine-tuned MLLMs (Sec. 4.2). These challenges highlight the gap between the potential of reasoning and its current limitations in DG, motivating the RD-MLDG framework introduced in the next section.

## 4.1 OPTIMIZATION GAP IN REASONING-CHAIN SUPERVISION

**Research Question.** *Does directly distilling class-relevant reasoning chains into MLLMs improve domain generalization in classification?*

**Experiment Setting.** As a case study, we conduct experiments on the TerraInc dataset (Beery et al., 2018), which contains 10 categories of wild animals across diverse camera locations, with domain shifts arising from viewpoint and background variations. We use InternVL3-8B (Chen et al., 2024) as the base model and compare two training regimes: (i) no-thinking (Li et al., 2025), where the model takes the image and input question (e.g., "What type of object is in this photo? Choose from the following options:") and directly predicts the label, and (ii) reasoning-chain supervision using DomainBed-Reasoning, where the same inputs are paired with multi-stage reasoning chains. Both regimes are evaluated under zero-shot prompting and supervised fine-tuning (SFT). To assess their behavior, we analyze token probability distributions and training dynamics (Fig. 3).

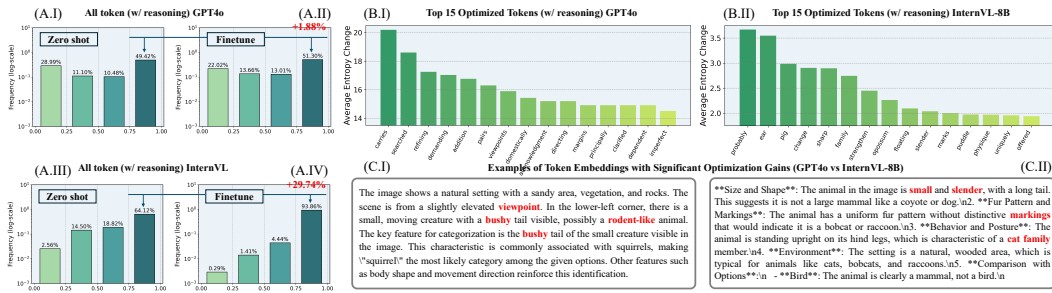

Figure 4: Illustration of Challenge 2 (see Sec. 4.2). (A) Token probability distributions on source-domain training data under zero-shot and fine-tuned settings: GPT-4o reasoning (A.I, A.II) vs InternVL-8B reasoning (A.III, A.IV). (B) Top-15 tokens with the largest entropy reduction during optimization, highlighting differences between GPT-4o (B.I) and InternVL-8B (B.II). (C) Qualitative examples corresponding to (B.I) and (B.II), with tokens receiving the strongest optimization gains highlighted in red.

**Observation.** First, in the zero-shot setting, adding reasoning improves classification performance under domain shift: InternVL3-8B shows a +43.28%p increase in ground-truth token probability compared to no-thinking predictions (Fig. 3.A, I vs III). Second, this benefit does not persist under supervised fine-tuning (SFT): reasoning-chain supervision yields slightly lower accuracy (–0.93%p) than direct label supervision (Fig. 3.A, II vs IV). Third, token-level distributions further highlight this optimization gap. No-thinking SFT shifts a large proportion of tokens from low to high probability (+43.38%p) (Fig. 3.B, I vs II), whereas reasoning-chain SFT produces only a marginal gain (+1.88%p) (Fig. 3.B, III vs IV). Training dynamics show a similar trend: no-thinking SFT fits classification tokens more effectively (92.23% vs 86.33% above the 75% confidence threshold), while reasoning-based SFT retains more low-probability samples (7.59% vs 2.05% below 25%) (Fig. 3.C.I–II). Finally, loss curves (Fig. 3.C.III) suggest that reasoning-based supervision converges more slowly, indicating that reasoning-chain supervision is harder to optimize than direct label prediction.

**Insight.** These results suggest that zero-shot reasoning can enhance out-of-domain generalization, but directly distilling reasoning chains during supervised fine-tuning is less effective than using label supervision alone. A key difficulty is that reasoning-chain supervision requires optimizing an intermediate step before predicting the final label, which slows convergence and reduces its effectiveness for classification learning. This limitation motivates the introduction of a direct classification pathway to guide reasoning supervision, making reasoning signals more effective for classification.

## 4.2 MISMATCHES IN REASONING PATTERNS ACROSS SOURCES

**Research Question.** *How do mismatches in reasoning patterns across different reasoning sources influence the supervision signals and optimization behavior of models?*

**Experiment Setting.** We use the same dataset and base model as in Sec. 4.1. To study mismatches in reasoning patterns, we compare fine-tuning with reasoning chains generated by a large MLLM (GPT-4o) versus those produced by the target model itself (InternVL3-8B). We then analyze how these different supervision sources affect token-level optimization by (i) comparing probability distributions of all tokens in zero-shot and fine-tuned settings, and (ii) examining the top-15 tokens with the largest entropy reduction (Fig. 4).

**Observation.** Fig. 4 shows how mismatches in reasoning patterns affect token-level optimization during fine-tuning. First, comparing token probability distributions (Fig. 4.A), reasoning chains generated by GPT-4o contain richer contextual descriptions (e.g., background, viewpoint), but fine-tuning with them yields only a marginal token shift (+1.88%p, A.I vs A.II). In contrast, when using reasoning chains generated by InternVL3-8B itself, fine-tuning produces a much larger token shift (+29.74%p, A.III vs A.IV), indicating that these chains are easier for the model to fit. Second, the entropy-based analysis (Fig. 4.B) reflects the same trend: GPT-4o reasoning exposes the model to fine-grained descriptive details that may be informative but are less directly aligned with classification, whereas self-generated reasoning emphasizes category-related tokens (e.g., 'cat family', 'bird'). Finally, qualitative examples (Fig. 4.C) illustrate these reasoning styles: GPT-4o produces longer, context-rich chains, while InternVL3-8B generates shorter, label-focused explanations. To-

kens highlighted in red denote those with the largest entropy reduction. Together, these observations suggest a trade-off: large-model reasoning provides richer but harder-to-optimize signals, while self-generated reasoning provides simpler cues that focus more directly on the category label.

**Insight.** These results suggest that mismatches between large-model reasoning (e.g., GPT-4o) and self-generated reasoning (InternVL3-8B) create a trade-off between contextual detail and ease of optimization. Large-model chains provide richer descriptions but include complex, high-entropy tokens that are harder to optimize, while self-generated chains are easier to fit but focus mainly on the category label and its immediate features. This trade-off may reduce the effectiveness of direct reasoning-chain supervision, motivating methods that preserve useful semantic detail while making reasoning signals easier to optimize.

## 5 METHOD

**Problem Definition.** Let $\mathcal{X}$ denote the input space and $\mathcal{Y}$ the label space. In domain generalization (DG), we assume access to $N_s$ labeled source domains $\hat{\mathcal{D}}^S = \{\hat{\mathcal{D}}_m^S\}_{m=1}^{N_s}$, where each $\hat{\mathcal{D}}_m^S$ is an empirical distribution over $\mathcal{X} \times \mathcal{Y}$ drawn from a distinct but related environment. For the $m$-th source domain with $N_m^s$ labeled samples, we write $\hat{\mathcal{D}}_m^S = \{(\mathbf{x}_i, y_i)\}_{i=1}^{N_m^s}$, where $\mathbf{x}_i \in \mathcal{X}$ denotes an input image and $y_i \in \mathcal{Y}$ its corresponding label. The goal of DG is to learn a model that generalizes to unseen target domains at test time, without access to any target data during training. The unseen target set is denoted as $\hat{\mathcal{D}}^T = \{\hat{\mathcal{D}}_m^T\}_{m=1}^{N_t}$, where each $\hat{\mathcal{D}}_m^T$ is a distribution over $\mathcal{X}$ corresponding to an unseen domain, with $\hat{\mathcal{D}}_m^T \notin \hat{\mathcal{D}}^S$ for all $m$.

**Overall Framework of our Method.** To address the challenges identified in Sec. 4, we propose RD-MLDG (Reasoning-Driven Multimodal LLM for Domain Generalization), a framework that integrates reasoning into DG. RD-MLDG has two modules: (i) Multi-Task Cross-Training (MTCT), which jointly optimizes a direct classification pathway and reasoning-chain supervision, so that the classification objective provides a stable signal to guide reasoning optimization, addressing the optimization difficulty in Challenge 1 (Sec. 4.1); and (ii) Self-Aligned Reasoning Regularization (SARR), which leverages self-labeling to produce reasoning chains consistent with the model's own reasoning style, thereby retaining useful contextual detail while reducing reasoning-pattern mismatches, addressing Challenge 2 (Sec. 4.2). Together, these modules make reasoning supervision more optimizable and informative, offering a structured way to incorporate reasoning into DG for improved out-of-domain performance.

### 5.1 MULTI-TASK CROSS-TRAINING

Motivated by the difficulty of optimizing reasoning-chain supervision observed in Challenge 1, Multi-Task Cross-Training (MTCT) trains the model to jointly optimize reasoning-chain supervision with a direct classification pathway. The key idea is that classification supervision serves as a simple and stable anchor objective, which guides reasoning optimization and helps alleviate the difficulty of fitting intermediate reasoning steps before predicting the final label. Concretely, for each training image $\mathbf{x}i$, we construct two prompts: (i) a **reasoning prompt** from DomainBed-Reasoning with its corresponding class-relevant reasoning chain $\mathbf{r}_i$, and (ii) a **classification prompt** (no-thinking) — "What type of object is in this photo? Choose from the following options:" — where the model selects from a candidate label list and the ground-truth label $y_i$ is used as the supervision signal. We denote these instruction templates as $\mathbf{q}_{\text{reason}}$ and $\mathbf{q}_{\text{cls}}$, respectively.

We fine-tune the base multimodal large language model (MLLM) by inserting LoRA adapters into both the vision encoder and the language decoder, enabling parameter-efficient supervised fine-tuning (SFT) without updating all parameters. Given a batch $\{(\mathbf{x}_i, y_i, \mathbf{r}_i)\}_{i=1}^B$, where $\mathbf{r}_i = (r_{i,1}, \ldots, r_{i,T_i})$ is the class-relevant reasoning chain for $\mathbf{x}_i$ and $\mathbf{r}_{i,<t} = (r_{i,1}, \ldots, r_{i,t-1})$ denotes the prefix sequence before predicting token $r_{i,t}$, the MTCT loss can be defined as follows:

$$\mathcal{L}_{\text{MTCT}} = \frac{1}{B} \sum_{i=1}^B \left[ -\log p_\theta\big(y_i \mid \mathbf{x}_i, \mathbf{q}_{\text{cls}}\big) - \frac{1}{T_i} \sum_{t=1}^{T_i} \log p_\theta\big(r_{i,t} \mid \mathbf{r}_{i,<t}, \mathbf{x}_i, \mathbf{q}_{\text{reason}}\big) \right], \quad (1)$$

where $T_i$ denotes the length (number of tokens) of the reasoning chain $\mathbf{r}_i$, and is used to normalize the generation loss so that longer chains do not dominate the gradients.

Table 1: Multi-source DG results (%) on DomainBed benchmark datasets. The best performances in comparisons are highlighted in **bold** and the second-best ones are marked with underlines.

| Method | Venue | PACS | VLCS | OfficeHome | TerraInc | Avg. |
|---|---|---|---|---|---|---|
| *ResNet-50 Based Method.* | | | | | | |
| CORAL (Sun & Saenko, 2016) | ICCV'16 | $86.20_{\pm 0.30}$ | $78.80_{\pm 0.60}$ | $68.70_{\pm 0.30}$ | $47.60_{\pm 1.00}$ | 70.33 |
| MLDG (Li et al., 2018a) | AAAI'18 | $84.90_{\pm 1.00}$ | $77.20_{\pm 0.40}$ | $66.80_{\pm 0.60}$ | $47.70_{\pm 0.90}$ | 69.15 |
| Mixstyle (Zhou et al., 2021) | ICLR'21 | $85.20_{\pm 0.30}$ | $77.90_{\pm 0.50}$ | $60.40_{\pm 0.20}$ | $44.00_{\pm 0.70}$ | 66.88 |
| DomainDrop (Guo et al., 2023a) | ICCV'23 | $87.90_{\pm 0.30}$ | $79.80_{\pm 0.30}$ | $68.70_{\pm 0.10}$ | $51.50_{\pm 0.40}$ | 71.98 |
| SMOS (Luo et al., 2024) | CVPR'24 | $89.40_{\pm 0.30}$ | $79.80_{\pm 0.10}$ | $71.60_{\pm 0.10}$ | $55.40_{\pm 0.40}$ | 74.05 |
| RES (Huang et al., 2025) | ECCV'24 | $90.00_{\pm 0.30}$ | $79.80_{\pm 0.20}$ | $71.80_{\pm 0.30}$ | $51.40_{\pm 0.60}$ | 73.25 |
| GGA (Ballas & Diou, 2025) | CVPR'25 | $87.30_{\pm 0.40}$ | $79.90_{\pm 0.40}$ | $68.50_{\pm 0.20}$ | $50.60_{\pm 0.10}$ | 71.58 |
| *VIT-B/16 Based Method.* | | | | | | |
| SWAD (Cha et al., 2021) | NIPS'21 | $91.30_{\pm 0.10}$ | $79.40_{\pm 0.40}$ | $76.90_{\pm 0.10}$ | $45.40_{\pm 0.50}$ | 73.25 |
| CLIP (Radford et al., 2021a) | - | $96.20_{\pm 0.10}$ | $81.70_{\pm 0.10}$ | $82.00_{\pm 0.10}$ | $33.40_{\pm 0.10}$ | 73.33 |
| CoOp (Zhou et al., 2022b) | IJCV'22 | $96.20_{\pm 0.10}$ | $77.60_{\pm 0.20}$ | $83.90_{\pm 0.10}$ | $48.80_{\pm 0.10}$ | 76.63 |
| MaPLe (Khattak et al., 2023) | CVPR'23 | $96.50_{\pm 0.20}$ | $82.20_{\pm 0.20}$ | $83.40_{\pm 0.00}$ | $50.20_{\pm 0.90}$ | 78.08 |
| SIMPLE$^+$ (Li et al., 2023b) | ICLR'23 | $\mathbf{99.00}_{\pm 0.10}$ | $82.70_{\pm 0.40}$ | $87.70_{\pm 0.40}$ | $59.00_{\pm 0.60}$ | 82.10 |
| VL2V-SD (Addepalli et al., 2024) | CVPR'24 | $95.67_{\pm 0.56}$ | $82.67_{\pm 0.36}$ | $85.44_{\pm 0.27}$ | $41.18_{\pm 0.74}$ | 76.24 |
| CLIP-LoRA Zanella & Ben Ayed (2024) | CVPR'24 | $97.10_{\pm 0.00}$ | $83.10_{\pm 0.00}$ | $83.90_{\pm 0.00}$ | $55.70_{\pm 0.00}$ | 79.95 |
| SPG (Bai et al., 2025) | ECCV'24 | $97.00_{\pm 0.50}$ | $82.40_{\pm 0.40}$ | $83.60_{\pm 0.40}$ | $50.20_{\pm 1.20}$ | 78.30 |
| DGCLDTP (Wen et al., 2025) | CVPR'25 | $97.03_{\pm 0.00}$ | $84.79_{\pm 0.00}$ | $87.65_{\pm 0.00}$ | $\underline{63.27}_{\pm 0.00}$ | 83.19 |
| *MLLM Based Method.* | | | | | | |
| GPT4o (OpenAI, 2023) | - | $97.83_{\pm 0.00}$ | $85.41_{\pm 0.00}$ | $\underline{90.12}_{\pm 0.00}$ | $60.49_{\pm 0.00}$ | $\underline{83.46}$ |
| InternVL3-8B (Zhu et al., 2025) | - | $96.26_{\pm 0.10}$ | $\underline{85.67}_{\pm 0.10}$ | $85.10_{\pm 0.10}$ | $46.84_{\pm 0.20}$ | 78.47 |
| InternVL3-8B + RD-MLDG (Ours) | - | $\underline{98.13}_{\pm 0.10}$ | $\mathbf{87.03}_{\pm 0.10}$ | $\mathbf{91.73}_{\pm 0.10}$ | $\mathbf{70.65}_{\pm 0.10}$ | $\mathbf{86.89}$ |

This formulation directly addresses the optimization difficulty identified in Sec. 4.1, making reasoning supervision more stable and effective for domain generalization.

## 5.2 Self-Aligned Reasoning Regularization

After the first stage MTCT training, the model has already been trained with reasoning chains distilled from large MLLMs (e.g., GPT-4o). However, directly relying on this supervision can be challenging due to reasoning-pattern mismatches, as discussed in Sec. 4.2. To mitigate this issue, we introduce Self-Aligned Reasoning Regularization (SARR), a soft self-labeling procedure. Specifically, the model generates its own reasoning under the instruction $\mathbf{q}_{\text{reason}}$. From each output, we extract the predicted conclusion enclosed between <CONCLUSION> and </CONCLUSION> and compare it with the ground-truth label $y_i$. Only reasoning chains with correct final conclusions are retained and used as refined supervision signals.

These retained reasoning chains are then combined with the classification prompt $\mathbf{q}_{\text{cls}}$ as part of the MTCT fine-tuning process. In each round of SARR, the reasoning chains, initially sourced from GPT-4o in the first stage, are replaced by self-generated reasoning chains in later stages. The training objective for each round of SARR remains the same as MTCT, where the model is fine-tuned using both reasoning chains and the classification prompt. Given a batch with self-generated reasoning chains $(\mathbf{x}_i, y_i, \hat{\mathbf{r}}_i)_{i=1}^{B'}$, where $\hat{\mathbf{r}}_i$ denotes a reasoning chain retained because its final conclusion matches the ground-truth label, the loss in SARR for each round can be defined as:

$$\mathcal{L}_{\text{SARR}} = \frac{1}{B'} \sum_{i=1}^{B'} \left[ -\log p_\theta(y_i \mid \mathbf{x}_i, \mathbf{q}_{\text{cls}}) - \frac{1}{T_i} \sum_{t=1}^{T_i} \log p_\theta(\hat{r}_{i,t} \mid \hat{\mathbf{r}}_{i,<t}, \mathbf{x}_i, \mathbf{q}_{\text{reason}}) \right]. \quad (2)$$

This self-labeling procedure can be repeated for $N$ rounds. In each round, the model is trained with both classification ($\mathbf{q}_{\text{cls}}$) and reasoning ($\mathbf{q}_{\text{reason}}$) prompts, generates reasoning chains for the source domain training data, and keeps only those whose final conclusions match the ground-truth label. The model is then fine-tuned again with these reasoning–classification pairs. Through this iterative refinement, the supervision becomes both informative and easier to fit, helping mitigate the reasoning-pattern mismatches identified in Sec. 4.2.

Table 2: Ablation study on each component on OfficeHome and TerraIncognita dataset.

| Idx | Method | OfficeHome | | | | | TerraIncognita | | | | |
| --- | --- | --- | --- | --- | --- | --- | --- | --- | --- | --- | --- |
| | | Art | Clipart | Product | Real | Avg. | L100 | L38 | L43 | L46 | Avg. |
| 1 | GPT-4o | 87.10 | 85.36 | 95.20 | 92.82 | 90.12 | 67.28 | 59.71 | 64.47 | 50.49 | 60.49 |
| 2 | InternVL3-2B | 3.79 | 3.64 | 2.70 | 3.49 | 3.41 | 0.16 | 0.25 | 0.15 | 0.33 | 0.22 |
| 3 | + CLS only | 88.59 | 81.49 | 94.46 | 92.10 | 89.16 | 75.26 | 64.66 | 69.98 | 55.99 | 66.47 |
| 4 | + Reasoning only (Baseline) | 87.03 | 82.13 | 93.77 | 91.45 | 88.60 | 74.79 | 61.26 | 69.40 | 58.56 | 66.00 |
| 5 | + MTCT | 89.00 | 83.71 | 95.49 | 93.53 | $90.43^{(\uparrow 1.83\%p)}$ | 77.82 | 65.03 | 72.71 | 59.41 | $68.74^{(\uparrow 2.74\%p)}$ |
| 6 | + SARR | 88.30 | 82.54 | 95.29 | 93.07 | $89.80^{(\uparrow 1.20\%p)}$ | 75.83 | 62.08 | 69.51 | 59.27 | $66.67^{(\uparrow 0.67\%p)}$ |
| 7 | + MTCT + SARR | 90.13 | 84.57 | 96.00 | 93.62 | $91.08^{(\uparrow 2.48\%p)}$ | 82.43 | 66.54 | 73.62 | 61.15 | $70.94^{(\uparrow 4.94\%p)}$ |
| 8 | InternVL3-8B | 81.95 | 78.26 | 91.91 | 88.27 | 85.10 | 61.84 | 41.27 | 45.60 | 38.66 | 46.84 |
| 9 | + CLS only | 87.27 | 82.59 | 95.11 | 92.59 | 89.39 | 72.80 | 65.64 | 69.70 | 58.62 | 66.69 |
| 10 | + Reasoning only (Baseline) | 86.59 | 82.45 | 94.33 | 91.66 | 88.76 | 74.42 | 60.39 | 68.21 | 55.21 | 64.56 |
| 11 | + MTCT | 89.33 | 83.53 | 95.79 | 93.67 | $90.58^{(\uparrow 1.81\%p)}$ | 76.78 | 60.55 | 71.94 | 59.47 | $67.19^{(\uparrow 2.63\%p)}$ |
| 12 | + SARR | 90.89 | 83.34 | 95.95 | 93.44 | $90.91^{(\uparrow 2.14\%p)}$ | 74.97 | 60.96 | 68.28 | 56.96 | $65.29^{(\uparrow 0.73\%p)}$ |
| 13 | + MTCT + SARR | 91.72 | 85.14 | 96.25 | 93.81 | $91.73^{(\uparrow 2.97\%p)}$ | 81.74 | 66.12 | 73.38 | 61.36 | $70.65^{(\uparrow 6.09\%p)}$ |

Figure 5: Analysis of MTCT (Sec. 5.1). Probability distributions on TerraInc source domain training data under direct label prediction, reasoning-only SFT, and MTCT SFT: (A) all tokens and (B) class tokens.

# 6 EXPERIMENTS

**Datasets and Evaluation Protocols.** We adopt four widely used benchmarks for evaluation: PACS (Li et al., 2017) (4 domains, 9,991 samples, 7 classes), VLCS (Li et al., 2017) (4 domains, 10,729 samples, 5 classes), OfficeHome (Venkateswara et al., 2017) (4 domains, 15,588 samples, 65 classes), and TerraInc (Beery et al., 2018) (4 domains, 24,788 samples, 10 classes).

For each dataset, we use the extended DomainBed-Reasoning version, where each image–label pair is paired with structured reasoning chains generated by GPT-4o (see Sec. 3). We evaluate classification under domain shift using the standard DomainBed leave-one-domain-out protocol and report the average test accuracy (Avg-acc) across all target domains. To reduce variance, results are averaged over three runs with different random seeds.

**Implementation Details.** We follow the supervised fine-tuning (SFT) procedure described in Sec. 5. Each training stage runs for 3 epochs with a batch size of 128 and a learning rate of 5e-4. We use AdamW as the optimizer. LoRA adapters (rank 8) are applied to both the vision encoder and the language decoder for parameter-efficient fine-tuning. For SARR (Sec. 5.2), we set the number of self-labeling rounds to $N = 3$. All experiments are conducted on 4 NVIDIA A100 GPUs (80GB).

## 6.1 RESULTS ON MULTIPLE DOMAIN GENERALIZATION

To demonstrate the effectiveness of our proposed RD-MLDG, we compare it with representative SOTA methods (Tab. 1). Built on InternVL3-8B, RD-MLDG achieves the best average accuracy of 86.89%, surpassing a strong commercial MLLM (GPT-4o, 83.46%) and competitive CLIP/ViT approaches (e.g., DGCLDTP, 83.19%). Per-dataset, RD-MLDG attains new best results on VLCS (87.03%), OfficeHome (91.73%), and TerraInc (70.65%), while remaining highly competitive on PACS (98.13%). To sum up, our proposed method has two main advantages: (1) Compared to feature-level invariant representation learning, RD-MLDG leverages class-relevant reasoning chains, which are explicitly observable and understandable by humans, rather than relying on latent feature representations that are difficult to interpret. (2) Through an efficient two-stage supervised fine-tuning procedure, our method avoids the need for complex training strategies while substantially enhancing the classification ability of the underlying MLLM under domain shift.

## 6.2 ABLATION STUDY

To further evaluate the effectiveness of each component, we conduct ablations using InternVL3-2B and InternVL3-8B on OfficeHome and TerraInc; results are shown in Tab. 2. The very low zero-shot

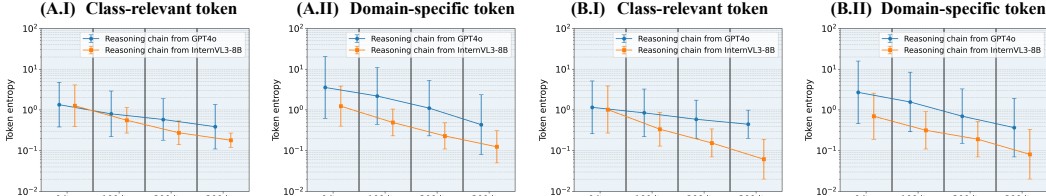

Figure 6: Analysis of SARR (see Sec. 5.2) from token entropy on TerraInc (A) and OfficeHome (B). We track the entropy of selected **class-relevant** (A.I & B.I) and **domain-specific** tokens (A.II & B.II) across fine-tuning iterations, with vertical bars denoting variation across samples. Blue curves correspond to fitting GPT-4o reasoning chains, and orange curves to InternVL3-8B self-generated reasoning chains.

result of InternVL3-2B ($2^{nd}$ row) is due to its **limited instruction-following** ability, but fine-tuning with reasoning chains supervision ($4^{th}$ row) quickly recovers strong baselines.

**Effectiveness of Multi-Task Cross-Training.** On both OfficeHome and TerraInc, direct label prediction outperforms reasoning-chain supervision: +0.56%p and +0.47%p on InternVL3-2B ($3^{rd}$ vs. $4^{th}$ rows), and +0.63%p and +2.13%p on InternVL3-8B ($9^{th}$ vs. $10^{th}$ rows), supporting the observation from Challenge 1 that reasoning-only supervision is less effective for classification. Relative to the reasoning-only baseline, MTCT yields additional improvements of +1.83%p and +2.74%p on InternVL3-2B ($4^{th}$ vs. $5^{th}$ rows), and +1.81%p and +2.63%p on InternVL3-8B ($10^{th}$ vs. $11^{th}$ rows), supporting that the direct classification pathway provides a stable signal that helps reasoning supervision contribute more effectively. Moreover, when combined with SARR, MTCT brings further gains of +1.28%p and +4.27%p on InternVL3-2B ($6^{th}$ vs. $7^{th}$ rows), and +0.83%p and +5.36%p on InternVL3-8B ($12^{th}$ vs. $13^{th}$ rows), reinforcing its role in guiding reasoning optimization.

Fig. 5 provides further evidence from the token-probability perspective. In Fig. 5.A, the **all-token distributions** show that MTCT still struggles to fully fit the semantically rich reasoning chains generated by GPT-4o, with 19.33% of tokens remaining at low probability after fine-tuning. By contrast, Fig. 5.B highlights consistent improvements in the **class-token distributions**: the proportion of high-confidence class tokens ($>0.75$) rises from 86.33% to 90.23%, while the share of low-confidence class tokens drops from 7.59% to 3.19%. These results indicate that MTCT does not substantially enhance the modeling of all reasoning details, but it does strengthen the fitting of class tokens, which directly supports the classification objective.

**Effectiveness of Self-Aligned Reasoning Regularization.** On OfficeHome and TerraInc, SARR improves accuracy over direct reasoning-chain supervision by +1.20%p and +0.67%p on InternVL3-2B ($4^{th}$ vs. $6^{th}$ rows), and by +2.14% and +0.73% on InternVL3-8B ($10^{th}$ vs. $12^{th}$ rows). When combined with MTCT, it brings further gains of +0.65%p and +2.20%p on InternVL3-2B ($5^{th}$ vs. $7^{th}$ rows), and +1.15%p and +3.46%p on InternVL3-8B ($11^{th}$ vs. $13^{th}$ rows). These results support the insight from Challenge 2: self-labeled reasoning retains class-relevant information from large-model supervision while producing signals that the model can fit more effectively.

Fig. 6.A reports token-entropy dynamics on TerraInc. We note two key observations. First, for **class-relevant tokens** (e.g., mark, floating), GPT-4o and InternVL3-8B reasoning chains start with similar entropy in the zero-shot stage, but as fine-tuning proceeds, entropy decreases faster for InternVL3-8B, showing that SARR shifts supervision toward signals the model can fit more readily. Second, for **domain-specific tokens** (e.g., demanding, viewpoint), GPT-4o reasoning chains have higher initial entropy than self-generated chains, reflecting richer but harder-to-fit details; with SARR, the model reduces fitting pressure on these tokens and instead focuses on class-relevant ones. This reallocation of optimization effort helps explain the accuracy gains of SARR. Results on OfficeHome (Fig. 6.B) show the same trend.

## 6.3 PARAMETER ANALYSIS

We further analyze how the number of self-labeling rounds $N$ affects performance on TerraInc (Fig. 7, top). We report $p$-values from paired t-tests, where runs with the same random seed are compared across adjacent $N$ settings. Accuracy shows a small but statistically significant improvement from 70.06% at $N = 1$ to 70.59% at $N = 2$ ($p < 0.01$). At $N = 3$, accuracy reaches 70.65%,

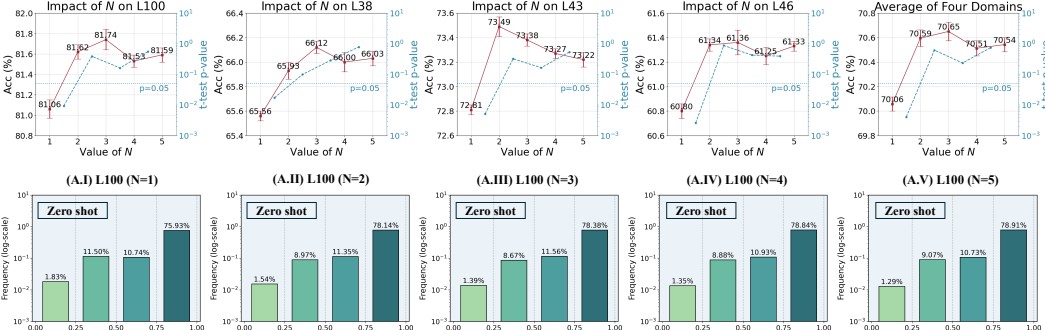

Figure 7: Impact of the number of self-labeling rounds $N$ in SARR (Sec. 5.2) on TerraInc using InternVL3-8B. Top: classification accuracy on four domains (L100, L38, L43, L46) and their average, with paired $t$-test $p$-values for adjacent $N$ shown on the right axis (reference line at $p = 0.05$). Bottom: zero-shot token probability distributions (log-scale) on L100, using supervision data from each self-labeling round.

but the improvement over $N = 2$ is not statistically significant ($p \approx 0.07$). For $N > 3$, accuracy remains stable between 70.50% and 70.60% with $p$-values above 0.1.

We also examine the zero-shot token-probability distributions across different self-labeling rounds (Fig. 7, bottom). From $N = 1$ to $N = 2$, the proportion of high-confidence tokens increases (2.27%) while low-confidence tokens decrease (0.29%), indicating a clearer supervision signal. For $N \geq 3$, the probability distribution remains nearly unchanged, showing that self-labeling converges within the first few rounds. We therefore set $N = 3$ in our experiments.

# 7 CONCLUSION

This work investigated how reasoning can contribute to domain generalization and identified two key challenges: optimization difficulty and reasoning-pattern mismatch. To address these issues, we proposed RD-MLDG, which integrates class-relevant reasoning chains with label supervision through MTCT and SARR. Experiments on DomainBed show that RD-MLDG achieves the best reported performance among strong baselines.

# 8 ACKNOWLEDGMENTS

This work was supported in part by the National Key R&D Program of China under Grant No.2023YFA1008600, in part by the National Natural Science Foundation of China under Grants 62576262, U22A2096, in part by the Key Research and Development Program of Shaanxi Province under grant 2024GX-YBXM135, 2024SF-YBXM-647, in part by the Fundamental Research Funds for the Central Universities under Grant QTZX25083, QTZX23042.

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

Table 3: Cross-domain divergence of visual embeddings and reasoning-chain embeddings for each TerraInc class.

| | bird | bobcat | cat | coyote | dog | empty | opossum | rabbit | raccoon | squirrel | Avg. |
|---|---|---|---|---|---|---|---|---|---|---|---|
| Domain divergence (Visual) | 0.209 | 0.251 | 0.314 | 0.387 | 0.213 | 0.144 | 0.266 | 0.167 | 0.258 | 0.176 | 0.239 |
| Domain divergence (Text) | 0.054 | 0.048 | 0.103 | 0.114 | 0.093 | 0.126 | 0.091 | 0.103 | 0.142 | 0.115 | 0.099 |

Zhipeng Xu, De Cheng, Xinyang Jiang, Nannan Wang, Dongsheng Li, and Xinbo Gao. Adversarial domain prompt tuning and generation for single domain generalization. In *Proceedings of the Computer Vision and Pattern Recognition Conference*, pp. 18584–18595, 2025.

Shen Yan, Huan Song, Nanxiang Li, Lincan Zou, and Liu Ren. Improve unsupervised domain adaptation with mixup training. *arXiv preprint arXiv:2001.00677*, 2020.

Zhihe Yang, Xufang Luo, Zilong Wang, Dongqi Han, Zhiyuan He, Dongsheng Li, and Yunjian Xu. Do not let low-probability tokens over-dominate in rl for llms. *arXiv preprint arXiv:2505.12929*, 2025.

Hantao Yao, Rui Zhang, and Changsheng Xu. Visual-language prompt tuning with knowledge-guided context optimization. In *Proceedings of the IEEE/CVF conference on computer vision and pattern recognition*, pp. 6757–6767, 2023.

Sangdoo Yun, Dongyoon Han, Seong Joon Oh, Sanghyuk Chun, Junsuk Choe, and Youngjoon Yoo. Cutmix: Regularization strategy to train strong classifiers with localizable features. In *Proceedings of the IEEE/CVF international conference on computer vision*, pp. 6023–6032, 2019.

Maxime Zanella and Ismail Ben Ayed. Low-rank few-shot adaptation of vision-language models. In *Proceedings of the IEEE/CVF Conference on Computer Vision and Pattern Recognition*, pp. 1593–1603, 2024.

Hongyi Zhang, Moustapha Cisse, Yann N Dauphin, and David Lopez-Paz. mixup: Beyond empirical risk minimization. In *International Conference on Learning Representations*, 2018.

Marvin Zhang, Henrik Marklund, Nikita Dhawan, Abhishek Gupta, Sergey Levine, and Chelsea Finn. Adaptive risk minimization: Learning to adapt to domain shift. *Advances in Neural Information Processing Systems*, 34:23664–23678, 2021.

Yuhui Zhang, Alyssa Unell, Xiaohan Wang, Dhruba Ghosh, Yuchang Su, Ludwig Schmidt, and Serena Yeung-Levy. Why are visually-grounded language models bad at image classification? *arXiv preprint arXiv:2405.18415*, 2024.

Kaiyang Zhou, Yongxin Yang, Yu Qiao, and Tao Xiang. Domain generalization with mixstyle. *arXiv preprint arXiv:2104.02008*, 2021.

Kaiyang Zhou, Jingkang Yang, Chen Change Loy, and Ziwei Liu. Conditional prompt learning for vision-language models. In *Proceedings of the IEEE/CVF conference on computer vision and pattern recognition*, pp. 16816–16825, 2022a.

Kaiyang Zhou, Jingkang Yang, Chen Change Loy, and Ziwei Liu. Learning to prompt for vision-language models. *International Journal of Computer Vision*, 130(9):2337–2348, 2022b.

Jinguo Zhu, Weiyun Wang, Zhe Chen, Zhaoyang Liu, Shenglong Ye, Lixin Gu, Hao Tian, Yuchen Duan, Weijie Su, Jie Shao, et al. Internvl3: Exploring advanced training and test-time recipes for open-source multimodal models. *arXiv preprint arXiv:2504.10479*, 2025.

## A  TRAINING PIPLINE FOR MCTC AND SARR MODULE

### A.1  QUANTITATIVE VALIDATION OF REASONING CHAIN DOMAIN INVARIANCE

To verify the hypothesis that reasoning chains are more domain-invariant than visual features, we performed a quantitative validation. Following existing work that measures cross-domain distributional discrepancies (Guo et al., 2023b), we compute the Maximum Mean Discrepancy (MMD)

between a source domain $\hat{\mathcal{D}}_m^S = \{(\mathbf{x}_i, y_i)\}_{i=1}^{N_m^s}$ and a target domain $\hat{\mathcal{D}}_{m'}^T = \{\mathbf{x}_j\}_{j=1}^{N_{m'}^t}$. Let $\mathbf{f}(\mathbf{x})$ denote the embedding extracted from CLIP's vision encoder for visual features, or from CLIP's text encoder for reasoning-chain embeddings. Under a linear kernel, MMD reduces to the squared distance between mean embeddings:

$$\mathrm{MMD}^2(\hat{\mathcal{D}}_m^S, \hat{\mathcal{D}}_{m'}^T) = \left\| \frac{1}{N_m^s} \sum_{i=1}^{N_m^s} \mathbf{f}(\mathbf{x}_i) - \frac{1}{N_{m'}^t} \sum_{j=1}^{N_{m'}^t} \mathbf{f}(\mathbf{x}_j) \right\|_2^2.$$

Using identical domain splits for both modalities, we compute the MMD for each category in TerraInc. As summarized in the Tab. 3, reasoning-chain embeddings exhibit dramatically lower cross-domain divergence than visual embeddings (average $0.239 \rightarrow 0.099$, a 58.6% reduction). This large gap provides clear quantitative evidence that reasoning chains capture semantically stable, class-relevant information that is far less sensitive to style, background, or environmental shifts than visual features. Therefore, the results strongly support our core hypothesis that reasoning chains are significantly more domain-invariant.

Table 4: Rejection rate across SARR iterations (N) on the four TerraInc domains.

|  | **L100** | **L38** | **L43** | **L38** | **Avg.** |
|---|---|---|---|---|---|
| N=0 | 41.78 | 39.25 | 40.84 | 36.18 | 39.51 |
| N=1 | 21.66 | 22.14 | 17.56 | 15.57 | 19.23 |
| N=2 | 18.49 | 17.98 | 15.07 | 13.64 | 16.30 |
| N=3 | 17.07 | 15.74 | 14.43 | 12.01 | 14.81 |

## A.2 REJECTION RATE ANALYSIS IN SARR ITERATIONS

To quantify the filtering behavior of SARR, we measure the rejection rate, defined as the percentage of self-generated reasoning chains whose <CONCLUSION> does not match the ground-truth label. Using InternVL3-8B on the TerraInc dataset (see Tab. 4), we observe that the rejection rate is relatively high at N = 0 (39.51%), reflecting the mismatch between GPT-4o–style reasoning and the model's native prediction tendencies. After the first SARR iteration, the rejection rate drops sharply to 19.23%, and continues to decrease at N = 2 and N = 3, eventually stabilizing around 15–17%.

This consistent downward trend indicates that, as SARR iterations progress, the model's self-generated reasoning chains increasingly lead to correct conclusions. Rather than performing random filtering, SARR progressively strengthens the semantic alignment between the reasoning chain and the model's final prediction. The improving correctness of self-generated chains demonstrates that the reasoning process itself becomes more stable, coherent, and predictive, providing direct evidence that RD-MLDG enhances reasoning quality rather than relying on incidental regularization effects.

Table 5: MTCT and SARR Performance on LLaVA-1.5 for TerraInc (%).

| Idx | Method | **TerraIncognita** | | | | |
|---|---|---|---|---|---|---|
| | | **L100** | **L38** | **L43** | **L46** | **Avg.** |
| 1 | InternVL3-2B + Reasoning only (Baseline) | 74.79 | 61.26 | 69.40 | 58.26 | 66.00 |
| 2 | **InternVL3-2B + MTCT** | 77.82 | 65.03 | 72.71 | 59.41 | 68.74$^{(\uparrow 2.74\%p)}$ |
| 3 | **InternVL3-2B + MTCT + SARR** | 82.43 | 66.54 | 73.62 | 61.15 | 70.94$^{(\uparrow 4.94\%p)}$ |
| 4 | InternVL3-8B + Reasoning only | 74.42 | 60.39 | 68.21 | 55.21 | 64.56 |
| 5 | **InternVL3-8B + MTCT** | 76.78 | 60.55 | 71.94 | 59.47 | 67.19$^{(\uparrow 2.63\%p)}$ |
| 6 | **InternVL3-8B + MTCT + SARR** | 81.74 | 66.12 | 73.38 | 61.36 | 70.65$^{(\uparrow 6.09\%p)}$ |
| 7 | LLaVA-1.5-7B + Reasoning only (Baseline) | 75.34 | 58.20 | 62.51 | 52.24 | 62.07 |
| 8 | **LLaVA-1.5-7B + MTCT** | 79.87 | 58.93 | 63.38 | 53.48 | 63.92$^{(\uparrow 1.85\%p)}$ |
| 9 | **LLaVA-1.5-7B + MTCT + SARR** | 80.93 | 60.01 | 64.51 | 55.14 | 65.15$^{(\uparrow 3.08\%p)}$ |

---

**Algorithm 1** RD-MLDG with MTCT and SARR

---

**Require:** Source domains $\{\hat{D}_m^S\}_{m=1}^{N_S}$, base model $f_\theta$, DomainBed-Reasoning dataset, self-labeling rounds $N$
**Ensure:** Fine-tuned model $f_\theta$

$\triangleright$ **Stage 1: Multi-Task Cross-Training (MTCT)**
1: **for** each minibatch $\{(x_i, y_i, r_i)\}_{i=1}^B$ **do**
2:     Construct classification prompt $q_{\text{cls}}$ and reasoning prompt $q_{\text{reason}}$
3:     Compute classification loss $L_{\text{cls}}$
4:     Compute reasoning loss $L_{\text{reason}}$
5:     Update $\theta$ with $L_{\text{MTCT}} = L_{\text{cls}} + L_{\text{reason}}$ (Eq. 1)
6: **end for**

$\triangleright$ **Stage 2: Self-Labeled Reasoning Regularization (SARR)**
7: **for** $k = 1$ to $N$ **do**
8:     **for** each sample $(x_i, y_i)$ **do**
9:         Generate reasoning $\hat{r}_i$ with $q_{\text{reason}}$
10:        Extract predicted conclusion between `<CONCLUSION>` and `</CONCLUSION>` tags
11:        **if** conclusion $= y_i$ **then**
12:            Retain $(x_i, y_i, \hat{r}_i)$ as supervision
13:        **end if**
14:     **end for**
15:     Fine-tune on retained pairs with $L_{\text{SARR}} = L_{\text{cls}} + L_{\text{reason}}$ (Eq. 2)
16: **end for**

---

## B   Results on smaller open-source model

To evaluate whether MTCT and SARR can generalize to smaller open-source models without dedicated reasoning pretraining, we conducted additional experiments on LLaVA-1.5-7B, whose reasoning ability is substantially weaker than that of InternVL3. This is directly reflected in the reasoning-only baselines (see Tab. 5): under the same setting on TerraInc, InternVL3-2B achieves an average accuracy of 66.00%, whereas LLaVA-1.5-7B reaches only 62.07%, indicating a clear gap in their initial reasoning capability. Despite this weaker backbone, **our method remains consistently effective**. As shown in the table, MTCT improves LLaVA-1.5-7B from 62.07% to 63.92%, and adding SARR further increases performance to 65.15%. A similar improvement is observed on InternVL3-2B, indicating that both components provide stable gains even when the underlying model lacks strong reasoning pretraining.

These results collectively demonstrate that RD-MLDG does not rely on a strong reasoning backbone. **Both MTCT and SARR transfer effectively to smaller, weaker, and fully open-source multimodal models**, showing that the proposed framework is broadly applicable beyond large reasoning-centric models.

## C   Results on Single Domain Generalization

**Background.** Most existing DG studies assume access to multiple source domains. However, in practice, models are often required to generalize from a single source domain to multiple unseen target domains (Single Domain Generalization, SDG). This setting is more challenging because the model cannot rely on inter-domain variation during training, making reasoning supervision potentially more valuable.

**Experiment Setting.** We conduct SDG experiments on the OfficeHome dataset, following the standard protocol where one domain (Art, Clipart, Product, Real World) is used as the source domain and the other three serve as target domains. We compare our method with representative SDG approaches, including domain-invariant representation learning (IRM, CORAL, VREx), data augmentation strategies (Mixup, CutMix, OrgMixup), adversarial learning (DANN, CDANN), and recent single-domain adaptation methods (RIDG, SAGM, PAPT). For our approach, we evaluate InternVL3-2B with RD-MLDG under the SDG setting.

Table 6: SDG results (%) on OfficeHome. One domain is used as the source domain and the others are used as the target domain. The best performances in comparisons are highlighted in **bold** and the second-best ones are marked with underlines.

| Method | Venue | A | C | P | R | Avg. |
|---|---|---|---|---|---|---|
| DANN (Ganin et al., 2016) | IJCAI'16 | 55.20 | 49.30 | 48.40 | 58.40 | 52.80 |
| CORAL (Sun & Saenko, 2016) | ICCV'16 | 55.60 | 52.80 | 50.30 | 59.40 | 54.50 |
| IRM (Arjovsky et al., 2019) | - | 54.90 | 53.20 | 48.60 | 59.20 | 54.00 |
| MMD (Li et al., 2018b) | ICCV'18 | 55.10 | 52.00 | 50.30 | 59.30 | 54.20 |
| OrgMixup (Zhang et al., 2018) | ICLR'18 | 56.00 | 54.40 | 50.40 | 61.00 | 55.50 |
| Mixup (Yan et al., 2020) | - | 55.50 | 54.10 | 49.40 | 59.40 | 54.60 |
| CutMix (Yun et al., 2019) | CVPR'19 | 53.50 | 52.20 | 47.70 | 60.20 | 53.40 |
| CDANN (Ganin et al., 2016) | - | 55.20 | 49.90 | 47.60 | 58.60 | 52.80 |
| GroupDRO (Sagawa et al., 2019) | ICLR'20 | 55.10 | 52.00 | 50.30 | 59.30 | 54.20 |
| MTL (Blanchard et al., 2021) | JMLR'21 | 55.30 | 53.30 | 49.00 | 60.40 | 54.50 |
| ARM (Zhang et al., 2021) | NeurIPS'21 | 55.00 | 51.60 | 47.30 | 59.30 | 53.30 |
| VREx (Krueger et al., 2021) | ICML'21 | 55.50 | 52.60 | 49.10 | 59.30 | 54.10 |
| Mixstyle (Zhou et al., 2021) | ICLR'21 | 44.30 | 29.80 | 33.60 | 48.50 | 39.00 |
| ERM (Vapnik, 2013) | ICLR'21 | 55.60 | 52.80 | 50.30 | 59.40 | 54.50 |
| SAM (Li et al., 2018b) | ICLR'21 | 56.90 | 53.80 | 50.90 | 61.50 | 55.80 |
| SagNet (Nam et al., 2021) | CVPR'21 | 56.90 | 53.40 | 50.80 | 61.20 | 55.60 |
| Fishr Rame et al. (2022) | ICML'22 | 55.10 | 51.20 | 49.20 | 59.90 | 53.90 |
| RIDG (Chen et al., 2023b) | ICCV'23 | 56.80 | 55.40 | 50.50 | 60.90 | 55.90 |
| SAGM (Wang et al., 2023) | CVPR'23 | 57.70 | 54.80 | 51.50 | 61.40 | 56.30 |
| ITTA (Chen et al., 2023a) | CVPR'23 | 56.00 | 51.50 | 50.50 | 61.60 | 54.90 |
| UDIM (Shin et al.) | ICLR'24 | 58.50 | 55.70 | 54.50 | 64.50 | 58.30 |
| PAPT (Xu et al., 2025) | CVPR'25 | 59.81 | 68.30 | 59.12 | 60.87 | 62.03 |
| **InternVL3-2B + RD-MLDG (Ours)** | - | **89.79** | **92.54** | **86.43** | **88.72** | **89.37** |

**Observation.** As shown in Tab. 6, RD-MLDG achieves a large margin over all baselines, reaching an average accuracy of 89.37%, compared to the best baseline PAPT at 62.03%. This indicates that reasoning supervision, even from a single domain, provides strong class-relevant signals that transfer effectively to unseen target domains. Notably, while prior methods rely on feature-level invariance or adversarial strategies, our approach explicitly incorporates reasoning chains, which appear more robust under severe domain shift.

Table 7: Base-to-new generalization results on FGVC-Aircraft.

| Idx | Method | FGVC-Aircraft | | |
|---|---|---|---|---|
| | | Base | New | H |
| 1 | InternVL3-8B | 20.35 | 17.83 | 19.01 |
| 2 | + Reasoning only (Baseline) | 39.20 | 15.44 | 22.11 |
| 3 | + MTCT | 56.42 | 18.60 | $27.95^{(\uparrow 5.84\%p)}$ |
| 4 | **+ MTCT + SARR** | **60.11** | **24.33** | $34.63^{(\uparrow 12.52\%p)}$ |

# D  RESULTS ON BASE-TO-NEW EXPERIMENT

To evaluate the broader advantages of reasoning-driven domain generalization, we perform a base-to-new class generalization experiment using the FGVC-Aircraft dataset (Maji et al., 2013).. Unlike the appearance-based shifts seen in the DomainBed benchmarks, the base-to-new protocol introduces semantic and compositional shifts within the label space. In this setting, the model must transfer reasoning from familiar categories to structurally distinct, unseen ones. This task is more challenging than traditional domain shifts as it requires the model to handle shifts in the underlying category structure rather than just appearance.

The FGVC-Aircraft dataset is particularly well-suited for this evaluation because its categories are defined by fine-grained structural and compositional attributes, such as wing geometry and engine configuration. These attributes make the task heavily reliant on structured reasoning rather than

Table 8: VQA and VE results (%) on VOLDOGER benchmark dataset. The best performances in comparisons are highlighted in **bold** and the second-best ones are marked with underlines.

| Method | VQA | | | | | VE | | | | |
|---|---|---|---|---|---|---|---|---|---|---|
| | Real | Cartoon | Pencil | Oil | Avg. | Real | Cartoon | Pencil | Oil | Avg. |
| ViT | 55.03 | 48.52 | 47.76 | 48.65 | 49.99 | 72.15 | 52.51 | 57.72 | 58.78 | 60.29 |
| CLIP (Radford et al., 2021a) | 58.23 | 49.11 | 50.41 | 49.41 | 51.79 | 73.10 | 55.85 | 61.61 | 60.93 | 62.87 |
| BLIP (Li et al., 2023a) | 59.19 | 50.29 | 51.32 | 50.88 | 52.92 | 66.73 | 48.26 | 52.93 | 53.62 | 55.39 |
| **Open-Source Models.** | | | | | | | | | | |
| BLIP2-FlanT5-XL (Li et al., 2023a) | 65.29 | 64.41 | 61.18 | 62.92 | 63.45 | 63.82 | 73.13 | 72.24 | 72.00 | 70.30 |
| PaliGemma (Steiner et al., 2024) | 80.59 | 79.41 | 75.29 | 75.59 | 77.72 | 33.43 | 33.91 | 35.02 | 34.79 | 34.29 |
| LLaVA-Next w/ Vicuna-7B (Liu et al., 2023) | 80.29 | 67.65 | 64.12 | 64.12 | 69.93 | 55.76 | 55.25 | 57.95 | 55.18 | 56.03 |
| LLaVA-Next w/ Mistral-7B (Liu et al., 2023) | 81.76 | 65.88 | 61.18 | 64.41 | 68.31 | 70.05 | 70.36 | 67.86 | 69.24 | 69.38 |
| **Proprietary Models.** | | | | | | | | | | |
| GPT-4-Vision 1106-preview (OpenAI, 2023) | 75.29 | 67.06 | 59.12 | 62.35 | 65.96 | 65.32 | 70.59 | 70.51 | 71.20 | 69.41 |
| GPT-4-Turbo 2024-04-09 (OpenAI, 2023) | 76.47 | 67.65 | 62.94 | 64.71 | 67.94 | 61.75 | 72.43 | 72.58 | 70.05 | 69.20 |
| GPT-4o 2024-05-13 (OpenAI, 2023) | 77.35 | **82.94** | **79.41** | **78.53** | **79.56** | 71.08 | 73.13 | 72.47 | 70.74 | 71.85 |
| Claude 3 Haiku | 75.00 | 67.35 | 62.06 | 62.35 | 66.69 | 58.18 | 63.55 | 67.86 | 66.47 | 64.01 |
| Claude 3 Sonnet | 68.24 | 74.12 | 72.35 | 70.29 | 71.25 | 59.22 | 72.28 | 72.24 | 71.08 | 68.70 |
| Claude 3 Opus | 63.53 | 63.82 | 61.76 | 63.24 | 63.09 | 59.91 | 66.65 | 61.18 | 64.06 | 62.95 |
| Gemini 1.0 Pro (Team et al., 2023) | 73.23 | 68.24 | 68.23 | 68.82 | 69.63 | 64.63 | 60.32 | 63.13 | 64.29 | 63.09 |
| Gemini 1.5 Flash (Team et al., 2024) | 75.88 | 78.82 | 73.82 | 72.94 | 75.36 | 64.17 | **74.39** | **73.96** | 72.35 | 71.22 |
| **RD-MLDG Method (Our).** | | | | | | | | | | |
| LLaVA-1.5-7B (Zero shot) | 0.29 | 0.59 | 0.29 | 0.29 | 0.37 | 0.23 | 0.12 | 0.23 | 0.23 | 0.20 |
| LLaVA-1.5-7B + Reasoning only (Baseline) | 75.88 | 59.41 | 59.71 | 59.71 | 63.68 | 67.51 | 66.32 | 67.28 | 67.35 | 67.12 |
| LLaVA-1.5-7B + MTCT | 78.53 | 68.82 | 64.71 | 65.59 | 69.41 | 72.35 | 65.97 | 67.74 | 66.24 | 68.08 |
| LLaVA-1.5-7B + MTCT + SARR | 80.88 | 71.17 | 67.06 | 67.35 | 71.62 | 73.96 | 69.12 | 68.89 | 68.54 | 70.13 |
| InternVL3-2B (Zero shot) | 0.59 | 0.29 | 0.29 | 0.59 | 0.44 | 0.34 | 0.23 | 0.34 | 0.34 | 0.31 |
| InternVL3-2B + Reasoning only (Baseline) | 74.12 | 66.76 | 66.18 | 63.82 | 67.72 | 67.97 | 63.67 | 65.55 | 67.17 | 66.09 |
| InternVL3-2B + MTCT | 83.53 | 69.41 | 68.53 | 69.12 | 72.65 | 71.20 | 65.05 | 66.94 | 66.71 | 67.48 |
| InternVL3-2B + MTCT + SARR | 85.29 | 71.47 | 69.71 | 71.17 | 74.41 | 74.30 | 68.20 | 69.35 | 68.66 | 70.13 |
| InternVL3-8B (Zero shot) | 81.76 | 69.12 | 69.71 | 68.24 | 72.21 | 70.85 | 62.28 | 65.90 | 64.86 | 65.97 |
| InternVL3-8B + Reasoning only (Baseline) | 81.47 | 74.12 | 70.88 | 67.94 | 73.60 | 73.50 | 70.13 | 68.32 | 70.39 | 70.59 |
| InternVL3-8B + MTCT | 83.82 | 77.94 | 71.47 | 68.53 | 75.44 | 73.96 | 68.54 | 69.35 | 71.57 | 70.62 |
| InternVL3-8B + MTCT + SARR | **85.00** | 81.76 | 72.94 | 69.41 | 77.28 | **74.76** | 70.85 | 70.62 | 72.11 | **72.09** |

surface-level visual cues, providing an ideal benchmark to test the model's ability to generalize through reasoning across different domains.

**Experiment setting:** Concretely, following existing work (Zhou et al., 2022b) that adopts the widely used base-to-new evaluation setting, we randomly split the 100 aircraft categories into 50 base and 50 new classes. All models are trained only on the base classes, and at test time we evaluate them on both base and unseen new classes, reporting performance on each split as well as their harmonic mean $H$. This setting ensures that good performance requires not only retaining accuracy on seen categories, but also transferring learned reasoning to semantically novel and compositionally distinct categories, thereby creating a genuine conceptual generalization scenario.

**Observation:** As shown in the Tab. 7, RD-MLDG improves harmonic mean accuracy from 19.01% → 34.63%, representing a substantial gain under this challenging conceptual shift. Notably, accuracy on unseen new classes increases from 15.44% → 24.33%, demonstrating that reasoning supervision enhances transfer to semantically novel and compositionally distinct categories rather than merely improving robustness to stylistic or appearance-level variations. These results provide direct empirical evidence that **RD-MLDG benefits conceptual generalization beyond the appearance-level shifts captured by existing benchmarks**.

# E EXPLORING THE EFFECTIVENESS OF RD-MLDG ON VQA AND VE (VISUAL ENTAILMENT)

**RD-MLDG is designed to be flexible and adaptable across various tasks.** Both MTCT and SARR are built on structured reasoning-chain supervision, making the framework task-agnostic. This means that as long as a task follows a general "vision → reasoning → output" pathway, RD-MLDG can be seamlessly applied without requiring changes to the model architecture or the optimization process.

In the Sec. 3, we adopt the DomainBed setting because it is the most widely used and standardized benchmark for studying domain generalization. Building on this foundation allows us to investigate reasoning-chain supervision within a well-established DG framework, ensuring that our analysis of the challenges introduced by reasoning in DG is performed under consistent and community-

recognized evaluation conditions. Through this setting, we identify and study two fundamental challenges caused by reasoning in DG: (i) the optimization difficulty and high-entropy gradients (Sec. 4.1), and (ii) the reasoning-pattern mismatch between external LLMs and target multimodal models (Sec. 4.2). MTCT and SARR are explicitly designed to address these two issues, and their mechanisms are independent of the specific downstream task used to study them.

**Extension to VQA and VE:** To verify the task generality of RD-MLDG beyond visual classification, we extend our method to two multimodal tasks: VQA and Visual Entailment (VE). We construct reasoning-augmented versions of both VOLDOGER-VQA and VOLDOGER-VE (Choi et al., 2025) so that their data formats align with DomainBed-Reasoning. Each sample is augmented with the same five structured components: `<SUMMARY>`, `<CAPTION>`, `<REASONING>`, `<REFLECTION>`, and `<CONCLUSION>`. This ensures that both MTCT and SARR can be applied without architectural changes to any model. Both datasets contain four visually heterogeneous domains (Real, Cartoon, Pencil, Oil). The style shifts significantly affect visual perception, textual grounding, and multimodal alignment. This makes VQA and VE natural testbeds for assessing whether reasoning-driven DG generalizes beyond classification.

**Observation:** Across both the VQA and VE tasks, our results (see Tab. 8) consistently demonstrate that RD-MLDG extends well beyond visual classification and remains effective in multimodal settings that require joint visual grounding and reasoning. On the VQA task, InternVL3-8B improves from 73.60% (reasoning-only baseline) to 75.44% with MTCT and further to 77.28% with SARR. On the VE task, the same model improves from 70.59% to 70.62% and finally to 72.09% after applying both modules. These gains are not isolated to a single model: LLaVA-1.5-7B, which lacks any reasoning-oriented pretraining, shows even larger improvements, rising from 63.68% → 69.41% → 71.62% on VQA and from 67.12% → 68.08% → 70.13% on VE. InternVL3-2B exhibits a similar trend on both tasks.

Beyond absolute gains, RD-MLDG also enhances the competitiveness of the underlying models relative to stronger multimodal LLMs. After applying MTCT and SARR, InternVL3-8B surpasses most zero-shot MLLMs, including BLIP2, the Claude 3 family, Gemini 1.0 and 1.5, and several LLaVA-Next variants, in average accuracy across all four visual styles for both VQA and VE. The fact that these improvements appear consistently across two distinct multimodal tasks, each involving substantial style-induced domain shifts (Real, Cartoon, Pencil, Oil), further confirms that MTCT and SARR enable the model to learn domain-invariant reasoning patterns that remain stable across different cross-modal decision-making objectives.

Taken together, the consistent performance gains across VQA and VE, coupled with improvements over a large set of open-source and proprietary multimodal baselines, provide strong evidence that RD-MLDG is a task-agnostic and broadly applicable reasoning-driven DG framework. The results confirm that our approach enhances the robustness and domain invariance of reasoning chains in diverse multimodal environments, thereby validating its generality far beyond visual classification.

# F  LARGE LANGUAGE MODEL USAGE STATEMENT

We relied on a large language model (LLM) only for language refinement (grammar, wording, and clarity). The LLM did not contribute to the conception of ideas, study design, data collection, analysis, or figure/table generation, and it did not write technical content. All methodological and experimental contributions, as well as the interpretation of results and final decisions, are by the authors.

---

**Instruct prompt:**

I have an image and a question that I want you to answer. I need you to strictly follow the format with five specific sections: SUMMARY, CAPTION, REASONING, REFLECTION, and CONCLUSION. It is crucial that you adhere to this structure exactly as outlined and that the final answer in the CONCLUSION matches the standard correct answer precisely. To explain further: In SUMMARY, briefly explain what steps you'll take to solve the problem. In CAPTION, describe the contents of the image, specifically focusing on details relevant to the question. In REASONING, outline a step-by-step thought process you would use to solve the problem based on the image. In REFLECTION, critically assess your reasoning, considering potential ambiguities, alternative interpretations, or justifying why your logic is most sound. In CONCLUSION, give the final answer in a direct format, and it must match the correct answer exactly. If it's a multiple choice question, the conclusion should only include the option without repeating what the option is. Here's how the format should look: <SUMMARY>[Summarize how you will approach the problem and explain the steps you will take to reach the answer.] </SUMMARY> <CAPTION>[Provide a detailed description of the image, particularly emphasizing the aspects related to the question.] </CAPTION> <REASONING>[Provide a chain-of-thought, logical explanation of the problem. This should outline step-by-step reasoning.] **</REASONING> <REFLECTION>[Review and evaluate your reasoning. Consider any uncertainties or edge cases, and reaffirm why the answer is reliable.] </REFLECTION>** <CONCLUSION>[State the final answer in a clear and direct format. It must match the correct answer exactly.] </CONCLUSION>(Do not forget </CONCLUSION>!) Please apply this updated format meticulously to analyze the given image and answer the related question, ensuring that the answer matches the standard one perfectly.

**Question prompt:**

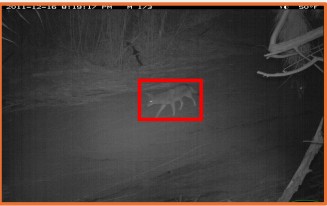

Question: Please describe the category of the image. I will provide a series of options, each separated by a comma.

Options: ['bird', 'bobcat', 'cat', 'coyote', 'dog', 'empty', 'opossum', 'rabbit', 'raccoon', 'squirrel']

---

Figure 8:  Input prompt template used for reasoning-chain generation. The prompt consists of two parts: (i) a question instruction paired with an image and candidate label options, and (ii) explicit guidelines requiring the model (GPT-4o) to output reasoning in a structured five-section format (SUMMARY, CAPTION, REASONING, REFLECTION, CONCLUSION). These generated reasoning chains are then paired with ground-truth labels and used to fine-tune downstream multimodal LLMs.

