# OpenReview forum: "Reasoning-Driven Multimodal LLM for Domain Generalization"
_ICLR.cc/2026/Conference — ICLR 2026 Poster_

### Official Review · Reviewer_jvkU · 2025-10-23

**Soundness:** 3
**Presentation:** 3
**Contribution:** 3
**Rating:** 6
**Confidence:** 4

**Summary:**

The paper introduces RD-MLDG, a framework that leverages reasoning chains in MLLMs to improve robustness under domain shift. The authors construct DomainBed-Reasoning, an extension of DomainBed in which each sample is paired with structured reasoning chains generated by GPT-4o. The authors propose two modules: Multi-Task Cross-Training (MTCT), which stabilizes reasoning-chain learning via an auxiliary classification pathway, and Self-Aligned Reasoning Regularization (SARR), which iteratively incorporates self-generated reasoning chains aligned with model predictions. Experiments on four DG benchmarks demonstrate state-of-the-art performance, with detailed ablations and analyses supporting the method’s contributions.

**Strengths:**

The paper makes good contribution by explicitly incorporating reasoning into domain generalization, an area where most prior methods have focused solely on feature-level invariance. The construction of DomainBed-Reasoning provides a valuable new benchmark for studying reasoning under domain shift, while the proposed RD-MLDG framework is empirically effective. The analysis of optimization gaps and reasoning mismatches is thorough and convincing, and the introduction of MTCT and SARR demonstrates clear performance improvements across multiple benchmarks.

**Weaknesses:**

- Dataset construction transparency: While DomainBed-Reasoning is central, the paper lacks sufficient details on reasoning-chain generation such as rejection sampling and filtering.
- Baseline fairness: Comparisons to prior DG methods and GPT-4o appear favorable, but it is unclear whether all baselines were fine-tuned under identical settings (prompt design, data splits, training budgets). For example, in Table 1, mixing zero-shot GPT-4o results with fine-tuned InternVL models may inflate apparent gains.
- Limited task scope: The study focuses exclusively on image classification. It remains uncertain whether the proposed framework generalizes to other multimodal DG tasks. Also although LoRA adapters are used, both MTCT and iterative SARR introduce added fine-tuning steps. Training efficiency and scalability to larger datasets are not fully addressed.

**Questions:**

- Could you provide more details on reasoning-chain generation, such as average number of reasoning chains generated per instance, rejection rate, and filtering criteria?
- Could you clarify in Table 1, for MLLM-based methods, which results are zero-shot and which are fine-tuning-based?
- Do you expect RD-MLDG to extend beyond image classification to other tasks? Additionally, given that preparing reasoning-chain supervision is non-trivial and both MTCT and iterative SARR introduce added fine-tuning steps, can you share your thoughts on the efficiency and scalability of the framework when applied to larger datasets?

---

> ### Author Response · Authors · 2025-11-22
>
> We sincerely thank the reviewer for the valuable questions. The Weaknesses \& Questions and Responses are as follows.
>
> $\textbf{Weakness 1 and Question 1:}$ Could you provide more details on reasoning-chain generation, such as average number of reasoning chains generated per instance, rejection rate, and filtering criteria?
>
> $\textbf{Response 1:}$ We thank the reviewer for the helpful question and address the three aspects as follows. 1) **Average number:** For each question–image pair, we generate 5 candidate reasoning chains using GPT4o; 2) **Rejection rate:** Among the 5 generated candidates, we retain only one reasoning chain per instance, specifically the first one that satisfies the filtering criteria described below; 3) **Filtering criteria:** A reasoning chain is accepted only if it contains all five required sections ($\texttt{\<SUMMARY\>}$, $\texttt{\<CAPTION\>}$, $\texttt{\<REASONING\>}$, $\texttt{\<REFLECTION\>}$, and $\texttt{\<CONCLUSION\>}$) and its $\texttt{\<CONCLUSION\>}$ matches the ground-truth label. Compared with LLaVA-CoT \[1\], our prompt additionally includes a $\texttt{\<REFLECTION\>}$ section, which encourages the model to self-check the intermediate reasoning and improves consistency between reasoning steps and the final conclusion. We further adopt the verification procedure from Appendix B of LLaVA-CoT (''Prompt for data verification'') to confirm that each retained $\texttt{\<CONCLUSION\>}$ is semantically aligned with its associated reasoning chain.
>
> \[1\] *Xu, Guowei, et al. "Llava-cot: Let vision language models reason step-by-step." Proceedings of the IEEE/CVF International Conference on Computer Vision. 2025.*
>
> $\textbf{Weakness 2 and Question 2 :}$ Could you clarify in Table 1, for MLLM-based methods, which results are zero-shot and which are fine-tuning-based?
>
> $\textbf{Response 2:}$ We thank the reviewer for pointing out the ambiguity in **Tab. 1 (main paper)**. To clarify, the row ''GPT-4o'' and the row ''InternVL3-8B'' both report zero-shot performance obtained directly from the pretrained models, without any fine-tuning on the proposed DomainBed-Reasoning dataset. In contrast, the row ''InternVL3-8B + RD-MLDG (Ours)'' corresponds to results after applying our MTCT+SARR fine-tuning procedure on the source domains of DomainBed-Reasoning. We will carefully revise the final version of the paper to make this distinction explicit in the **Tab. 1 (main paper)** caption.

---

> > ### Author Response · Authors · 2025-11-22
> >
> > $\textbf{Weakness 3 and Question 3:}$ Do you expect RD-MLDG to extend beyond image classification to other tasks? Additionally, given that preparing reasoning-chain supervision is non-trivial and both MTCT and iterative SARR introduce added fine-tuning steps, can you share your thoughts on the efficiency and scalability of the framework when applied to larger datasets?
> >
> > **Table R.1: VQA and VE experiment**
> > | Method | VQA Real | VQA Cartoon | VQA Pencil | VQA Oil | VQA Avg | VE Real | VE Cartoon | VE Pencil | VE Oil | VE Avg |
> > |:------|:--------:|:-----------:|:----------:|:-------:|:-------:|:-------:|:----------:|:---------:|:------:|:------:|
> > | ViT | 55.03 | 48.52 | 47.76 | 48.65 | 49.99 | 72.15 | 52.51 | 57.72 | 58.78 | 60.29 |
> > | CLIP | 58.23 | 49.11 | 50.41 | 49.41 | 51.79 | 73.10 | 55.85 | 61.61 | 60.93 | 62.87 |
> > | BLIP | 59.19 | 50.29 | 51.32 | 50.88 | 52.92 | 66.73 | 48.26 | 52.93 | 53.62 | 55.39 |
> > | **Open-Source Models** |  |  |  |  |  |  |  |  |  |  |
> > | BLIP2-FlanT5-XL | 65.29 | 64.41 | 61.18 | 62.92 | 63.45 | 63.82 | 73.13 | 72.24 | 72.00 | 70.30 |
> > | PaliGemma | 80.59 | 79.41 | 75.29 | 75.59 | 77.72 | 33.43 | 33.91 | 35.02 | 34.79 | 34.29 |
> > | LLaVA-Next w/ Vicuna-7B | 80.29 | 67.65 | 64.12 | 64.12 | 69.93 | 55.76 | 55.25 | 57.95 | 55.18 | 56.03 |
> > | LLaVA-Next w/ Mistral-7B | 81.76 | 65.88 | 61.18 | 64.41 | 68.31 | 70.05 | 70.36 | 67.86 | 69.24 | 69.38 |
> > | **Proprietary Models** |  |  |  |  |  |  |  |  |  |  |
> > | GPT-4-Vision 1106-preview | 75.29 | 67.06 | 59.12 | 62.35 | 65.96 | 65.32 | 70.59 | 70.51 | 71.20 | 69.41 |
> > | GPT-4-Turbo 2024-04-09 | 76.47 | 67.65 | 62.94 | 64.71 | 67.94 | 61.75 | 72.43 | 72.58 | 70.05 | 69.20 |
> > | GPT-4o 2024-05-13 | 77.35 | 82.94 | 79.41 | 78.53 | 79.56 | 71.08 | 73.13 | 72.47 | 70.74 | 71.85 |
> > | Claude 3 Haiku | 75.00 | 67.35 | 62.06 | 62.35 | 66.69 | 58.18 | 63.55 | 67.86 | 66.47 | 64.01 |
> > | Claude 3 Sonnet | 68.24 | 74.12 | 72.35 | 70.29 | 71.25 | 59.22 | 72.28 | 72.24 | 71.08 | 68.70 |
> > | Claude 3 Opus | 63.53 | 63.82 | 61.76 | 63.24 | 63.09 | 59.91 | 66.65 | 61.18 | 64.06 | 62.95 |
> > | Gemini 1.0 Pro | 73.23 | 68.24 | 68.23 | 68.82 | 69.63 | 64.63 | 60.32 | 63.13 | 64.29 | 63.09 |
> > | Gemini 1.5 Flash | 75.88 | 78.82 | 73.82 | 72.94 | 75.36 | 64.17 | 74.39 | 73.96 | 72.35 | 71.22 |
> > | **RD-MLDG Method (Ours)** |  |  |  |  |  |  |  |  |  |  |
> > | LLaVA-1.5-7B (Zero shot) | 0.29 | 0.59 | 0.29 | 0.29 | 0.37 | 0.23 | 0.12 | 0.23 | 0.23 | 0.20 |
> > | LLaVA-1.5-7B + Reasoning only | 75.88 | 59.41 | 59.71 | 59.71 | 63.68 | 67.51 | 66.32 | 67.28 | 67.35 | 67.12 |
> > | LLaVA-1.5-7B + MTCT | 78.53 | 68.82 | 64.71 | 65.59 | 69.41 | 72.35 | 65.97 | 67.74 | 66.24 | 68.08 |
> > | LLaVA-1.5-7B + MTCT + SARR | 80.88 | 71.17 | 67.06 | 67.35 | 71.62 | 73.96 | 69.12 | 68.89 | 68.54 | 70.13 |
> > | InternVL3-2B (Zero shot) | 0.59 | 0.29 | 0.29 | 0.59 | 0.44 | 0.34 | 0.23 | 0.34 | 0.34 | 0.31 |
> > | InternVL3-2B + Reasoning only | 74.12 | 66.76 | 66.18 | 63.82 | 67.72 | 67.97 | 63.67 | 65.55 | 67.17 | 66.09 |
> > | InternVL3-2B + MTCT | 83.53 | 69.41 | 68.53 | 69.12 | 72.65 | 71.20 | 65.05 | 66.94 | 66.71 | 67.48 |
> > | InternVL3-2B + MTCT + SARR | 85.29 | 71.47 | 69.71 | 71.17 | 74.41 | 74.30 | 68.20 | 69.35 | 68.66 | 70.13 |
> > | InternVL3-8B (Zero shot) | 81.76 | 69.12 | 69.71 | 68.24 | 72.21 | 70.85 | 62.28 | 65.90 | 64.86 | 65.97 |
> > | InternVL3-8B + Reasoning only | 81.47 | 74.12 | 70.88 | 67.94 | 73.60 | 73.50 | 70.13 | 68.32 | 70.39 | 70.59 |
> > | InternVL3-8B + MTCT | 83.82 | 77.94 | 71.47 | 68.53 | 75.44 | 73.96 | 68.54 | 69.35 | 71.57 | 70.62 |
> > | InternVL3-8B + MTCT + SARR | 85.00 | 81.76 | 72.94 | 69.41 | 77.28 | 74.76 | 70.85 | 70.62 | 72.11 | 72.09 |

---

> > > ### Author Response · Authors · 2025-11-22
> > >
> > > $\textbf{Response 3:}$ We thank the reviewer for the helpful question and address the two aspects as follows:
> > >
> > > **(1) Extension beyond image classification:** Yes. RD-MLDG is fundamentally a task-agnostic framework because both MTCT and SARR operate on structured reasoning-chain supervision rather than any task-specific architecture. The mechanisms introduced in RD-MLDG, which reduce the optimization difficulty of reasoning-chain training and mitigate the reasoning-pattern mismatch between external LLMs and multimodal models, apply to any task that follows a ''vision → reasoning → output'' computational pathway. The classification experiments in the main paper were chosen because DomainBed provides a widely adopted and standardized evaluation environment for analyzing these mechanisms, rather than because of any limitation of the method itself.
> > >
> > > To further demonstrate task generality, we extend RD-MLDG to two multimodal tasks, VQA and Visual Entailment (VE), by constructing reasoning-augmented versions of the VOLDOGER-VQA and VOLDOGER-VE \[1\] datasets. Each sample is enriched with a $\texttt{\<SUMMARY\>}$, $\texttt{\<CAPTION\>}$, $\texttt{\<REASONING\>}$, $\texttt{\<REFLECTION\>}$, and $\texttt{\<CONCLUSION\>}$ to align the data format with DomainBed-Reasoning. This allows MTCT and SARR to be applied directly without modifying the underlying model architectures.
> > >
> > > Across both tasks, RD-MLDG yields consistent out-of-domain improvements (see **Tab. R.1**). On VQA, LLaVA-1.5-7B improves from 63.68\% to 69.41\% and then to 71.62\% after applying MTCT and SARR, even though the model does not include any reasoning-oriented pretraining. InternVL models show the same trend. InternVL3-8B improves from 73.60\% to 75.44\% and then to 77.28\%. On VE, similar improvements are observed. LLaVA-1.5-7B improves from 67.12\% to 68.08\% and then to 70.13\%. InternVL3-8B improves from 70.59\% to 70.62\% and then to 72.09\%. These results demonstrate that RD-MLDG transfers effectively to varied multimodal reasoning tasks and consistently mitigates distribution shift beyond image classification.
> > >
> > > **Table R.2: Ablation on SARR efficiency: varying the number of iterations (N) and epochs per iteration**
> > > | Setting        | Location 100 | Location 38 | Location 43 | Location 46 | Avg. |
> > > |:--------------|:------------:|:-----------:|:-----------:|:-----------:|:----:|
> > > | N=1, epoch=3   | 81.06        | 65.56       | 72.81       | 60.80       | 70.06 |
> > > | N=3, epoch=1   | 81.50        | 65.71       | 73.00       | 61.32       | 70.39 |
> > > | N=3, epoch=3   | 81.74        | 66.12       | 73.38       | 61.36       | 70.65 |
> > >
> > > **(2) Training efficiency and scalability:** For MTCT, an additional classification objective is added to the reasoning-chain supervision. Compared with training LLM  with only reasoning chains as supervision, adding an extra classification objective introduce negligible overhead, because the direct classification pathway is required to fit far fewer tokens compared to the long reasoning chain.
> > >
> > > For SARR, once GPT-4o reasoning chains are replaced by self-labeled chains, the model no longer needs to handle cross-model reasoning mismatches, making optimization substantially easier. As a result, our experiments show that even under constrained computational budgets, such as fewer SARR rounds N or reduced epochs per round, the proposed approach consistently delivers notable performance gains. Specifically, we evaluated a lightweight setting that uses only one epoch per SARR round while keeping N = 3 (see **Tab. R.2**). This reduces the total fine-tuning cost to nearly that of a single full SARR round, yet the performance remains close to the original three-epoch configuration. After the first SARR round, supervision fully switches to the model’s own reasoning chains, and this alignment allows each subsequent round to converge efficiently within a single epoch.
> > >
> > > As a result, both MTCT and SARR do not introduce substantial additional computational cost compared with standard supervised fine-tuning, and thus the training efficiency and scalability of RD-MLDG can be well maintained.
> > >
> > > \[1\] *Choi, Juhwan, et al. "Voldoger: Llm-assisted datasets for domain generalization in vision-language tasks." Proceedings of the IEEE/CVF International Conference on Computer Vision. 2025.*

---

### Official Review · Reviewer_dvM6 · 2025-10-29

**Soundness:** 2
**Presentation:** 2
**Contribution:** 2
**Rating:** 2
**Confidence:** 3

**Summary:**

This paper proposes **RD-MLDG**, a *reasoning-driven multimodal large language model (MLLM)* framework designed to improve **domain generalization (DG)** — the ability of models to perform well on unseen domains. Unlike prior DG methods that focus primarily on learning invariant visual representations, this work leverages **reasoning chains** that explicitly describe class-relevant visual evidence to encourage *process-level* generalization.

To support this, the authors introduce **DomainBed-Reasoning**, an extension of the DomainBed benchmark that pairs each image with structured reasoning chains generated by GPT-4o. Through empirical analyses, they identify two key challenges:
1. **Optimization difficulty** — supervising models with reasoning chains is harder than optimizing for direct label prediction.
2. **Reasoning-pattern mismatch** — the reasoning patterns generated by large language models differ from those produced by the fine-tuned model.

To address these issues, the paper introduces two components:
* **MTCT (Multi-Task Cross-Training):** jointly optimizes classification and reasoning objectives to stabilize training.
* **SARR (Self-Aligned Reasoning Regularization):** a self-labeling approach that aligns the model’s reasoning style while maintaining semantic richness.

Experiments on PACS, VLCS, OfficeHome, and TerraIncognita demonstrate that RD-MLDG achieves **state-of-the-art performance** (average 86.89%), outperforming prior DG methods based on non-MLLM architectures.

**Strengths:**

* **Clear problem identification and analysis:** The authors systematically diagnose optimization and reasoning-pattern gaps through quantitative studies (e.g., token probability and entropy analysis).
* **(Minor) Dataset contribution:** DomainBed-Reasoning provides a useful benchmark for reasoning-based domain generalization and may foster further research, even if its construction is relatively straightforward.

**Weaknesses:**

* **Limited methodological novelty:** The distinction between MTCT and SARR losses is unclear, as they appear almost identical. In particular, the novelty of SARR seems limited, resembling a form of rejection sampling.
* **Evaluation scope:** The experiments focus mainly on visual classification tasks; it remains unclear whether reasoning-driven DG generalizes to other multimodal tasks (e.g., VQA, image-text retrieval).
* **Baseline coverage:** Comparisons are mainly against non-MLLM DG methods, making it difficult to assess the true advantage of RD-MLDG within the MLLM landscape.
* **(Minor) Writing:** The introduction could benefit from citations to recent survey papers summarizing reasoning-based or MLLM-based DG approaches.

**Questions:**

- Line 053: Please ensure the sentence ends with a period.
- Line 318: The description of how reasoning chains are combined with the classification prompt is unclear.

---

> ### Author Response · Authors · 2025-11-22
>
> We sincerely thank the reviewer for the valuable questions. The Weaknesses \& Questions and Responses are as follows.
>
> $\textbf{Weakness 1:}$ Limited methodological novelty: The distinction between MTCT and SARR losses is unclear, as they appear almost identical. In particular, the novelty of SARR seems limited, resembling a form of rejection sampling.
>
> $\textbf{Response 1:}$ We respectfully disagree with the reviewer’s claim that ''the distinction between MTCT and SARR losses is unclear'' and that ''SARR resembles a form of rejection sampling.'' These comments stem from a misunderstanding of how MTCT and SARR function within our framework. To clarify this, we explain the distinction from two complementary perspectives.
>
> **(1) Distinction between MTCT and SARR.** The reviewer's concern arises from a misunderstanding of the relationship between MTCT and SARR in our method.
> **MTCT and SARR are not two parallel modules with separate loss functions**. MTCT is the inner training objective used within each fine-tuning stage, while SARR is an outer multi-round self-alignment framework. In each SARR round, the supervision distribution is updated, and the model is fine-tuned again using MTCT on the updated data. In other words, each SARR round internally runs MTCT once. Ignoring this inner–outer hierarchy leads to the incorrect impression that MTCT and SARR are ''two identical losses.''
>
> Given this structure, the roles and functional logic of MTCT and SARR are fundamentally different:
> * **MTCT** focuses on improving the optimizability of reasoning-chain supervision. Reasoning chains are long, high-entropy sequences and produce unstable gradients—this is Challenge 1 (Sec. 4.1). MTCT jointly optimizes a low-entropy classification objective and a high-entropy reasoning objective. The classification pathway provides a stable optimization anchor, enabling the model to learn class-relevant information from complex reasoning chains and significantly stabilizing token-level learning dynamics (**Fig. 5 (main paper)** and **Tab. 2 (main paper)**). Thus, MTCT answers the question: ''How can we optimize reasoning-chain supervision effectively?''
> * **SARR**, in contrast, focuses on making the supervision distribution itself learnable. As shown in Sec. 4.2, GPT-4o's reasoning style (length, structure, detail density) mismatches InternVL's natural reasoning patterns, and this mismatch fundamentally limits optimizability, which corresponds to Challenge 2. SARR addresses this by iteratively performing: (i) generating InternVL's own reasoning chains, (ii) filtering for correctness, and (iii) fine-tuning using MTCT on this self-aligned supervision. Through these rounds, the supervision distribution gradually shifts from GPT-4o's style to InternVL's own reasoning style, reduces entropy, and becomes easier to optimize. In essence, SARR answers the question: ''What form should the supervision take so that the model can learn it effectively?''
>
> **(2) Difference between SARR and rejection sampling.** The reviewer's comparison between SARR and rejection sampling is based on a misunderstanding. These two procedures serve entirely different purposes.
>
> * **Rejection sampling** is a data curation technique aimed at filtering annotations to improve dataset quality. It operates entirely at the data level and does not influence training dynamics or alter the supervision distribution during optimization.
> * **SARR**, in contrast, is a **training framework** designed to transform supervision through self-alignment. It involves an iterative cycle of: (i) generating model-specific reasoning with model trained in previous round, (ii) filtering generated results with correct prediction, and (iii) fine-tuning current round's model with MTCT using the generated self-aligned reasoning distribution.
>
> This process addresses the cross-model reasoning mismatch, a fundamental barrier to optimizability, by progressively aligning supervision with the model's own reasoning style. It is therefore not a data-filtering step, but a principled training mechanism for reasoning style transformation. The improvements shown in **Tab. 2 (lines 5–7 and 11–13, main paper)** cannot be explained by simple data filtering; they result from the dynamic self-alignment process unique to SARR.

---

> > ### Author Response · Authors · 2025-11-22
> >
> > $\textbf{Weakness 2:}$ Evaluation scope: The experiments focus mainly on visual classification tasks; it remains unclear whether reasoning-driven DG generalizes to other multimodal tasks (e.g., VQA, image-text retrieval).
> >
> > **Table R.1: VQA and VE experiment**
> > | Method | VQA Real | VQA Cartoon | VQA Pencil | VQA Oil | VQA Avg | VE Real | VE Cartoon | VE Pencil | VE Oil | VE Avg |
> > |:------|:--------:|:-----------:|:----------:|:-------:|:-------:|:-------:|:----------:|:---------:|:------:|:------:|
> > | ViT | 55.03 | 48.52 | 47.76 | 48.65 | 49.99 | 72.15 | 52.51 | 57.72 | 58.78 | 60.29 |
> > | CLIP | 58.23 | 49.11 | 50.41 | 49.41 | 51.79 | 73.10 | 55.85 | 61.61 | 60.93 | 62.87 |
> > | BLIP | 59.19 | 50.29 | 51.32 | 50.88 | 52.92 | 66.73 | 48.26 | 52.93 | 53.62 | 55.39 |
> > | **Open-Source Models** |  |  |  |  |  |  |  |  |  |  |
> > | BLIP2-FlanT5-XL | 65.29 | 64.41 | 61.18 | 62.92 | 63.45 | 63.82 | 73.13 | 72.24 | 72.00 | 70.30 |
> > | PaliGemma | 80.59 | 79.41 | 75.29 | 75.59 | 77.72 | 33.43 | 33.91 | 35.02 | 34.79 | 34.29 |
> > | LLaVA-Next w/ Vicuna-7B | 80.29 | 67.65 | 64.12 | 64.12 | 69.93 | 55.76 | 55.25 | 57.95 | 55.18 | 56.03 |
> > | LLaVA-Next w/ Mistral-7B | 81.76 | 65.88 | 61.18 | 64.41 | 68.31 | 70.05 | 70.36 | 67.86 | 69.24 | 69.38 |
> > | **Proprietary Models** |  |  |  |  |  |  |  |  |  |  |
> > | GPT-4-Vision 1106-preview | 75.29 | 67.06 | 59.12 | 62.35 | 65.96 | 65.32 | 70.59 | 70.51 | 71.20 | 69.41 |
> > | GPT-4-Turbo 2024-04-09 | 76.47 | 67.65 | 62.94 | 64.71 | 67.94 | 61.75 | 72.43 | 72.58 | 70.05 | 69.20 |
> > | GPT-4o 2024-05-13 | 77.35 | 82.94 | 79.41 | 78.53 | 79.56 | 71.08 | 73.13 | 72.47 | 70.74 | 71.85 |
> > | Claude 3 Haiku | 75.00 | 67.35 | 62.06 | 62.35 | 66.69 | 58.18 | 63.55 | 67.86 | 66.47 | 64.01 |
> > | Claude 3 Sonnet | 68.24 | 74.12 | 72.35 | 70.29 | 71.25 | 59.22 | 72.28 | 72.24 | 71.08 | 68.70 |
> > | Claude 3 Opus | 63.53 | 63.82 | 61.76 | 63.24 | 63.09 | 59.91 | 66.65 | 61.18 | 64.06 | 62.95 |
> > | Gemini 1.0 Pro | 73.23 | 68.24 | 68.23 | 68.82 | 69.63 | 64.63 | 60.32 | 63.13 | 64.29 | 63.09 |
> > | Gemini 1.5 Flash | 75.88 | 78.82 | 73.82 | 72.94 | 75.36 | 64.17 | 74.39 | 73.96 | 72.35 | 71.22 |
> > | **RD-MLDG Method (Ours)** |  |  |  |  |  |  |  |  |  |  |
> > | LLaVA-1.5-7B (Zero shot) | 0.29 | 0.59 | 0.29 | 0.29 | 0.37 | 0.23 | 0.12 | 0.23 | 0.23 | 0.20 |
> > | LLaVA-1.5-7B + Reasoning only | 75.88 | 59.41 | 59.71 | 59.71 | 63.68 | 67.51 | 66.32 | 67.28 | 67.35 | 67.12 |
> > | LLaVA-1.5-7B + MTCT | 78.53 | 68.82 | 64.71 | 65.59 | 69.41 | 72.35 | 65.97 | 67.74 | 66.24 | 68.08 |
> > | LLaVA-1.5-7B + MTCT + SARR | 80.88 | 71.17 | 67.06 | 67.35 | 71.62 | 73.96 | 69.12 | 68.89 | 68.54 | 70.13 |
> > | InternVL3-2B (Zero shot) | 0.59 | 0.29 | 0.29 | 0.59 | 0.44 | 0.34 | 0.23 | 0.34 | 0.34 | 0.31 |
> > | InternVL3-2B + Reasoning only | 74.12 | 66.76 | 66.18 | 63.82 | 67.72 | 67.97 | 63.67 | 65.55 | 67.17 | 66.09 |
> > | InternVL3-2B + MTCT | 83.53 | 69.41 | 68.53 | 69.12 | 72.65 | 71.20 | 65.05 | 66.94 | 66.71 | 67.48 |
> > | InternVL3-2B + MTCT + SARR | 85.29 | 71.47 | 69.71 | 71.17 | 74.41 | 74.30 | 68.20 | 69.35 | 68.66 | 70.13 |
> > | InternVL3-8B (Zero shot) | 81.76 | 69.12 | 69.71 | 68.24 | 72.21 | 70.85 | 62.28 | 65.90 | 64.86 | 65.97 |
> > | InternVL3-8B + Reasoning only | 81.47 | 74.12 | 70.88 | 67.94 | 73.60 | 73.50 | 70.13 | 68.32 | 70.39 | 70.59 |
> > | InternVL3-8B + MTCT | 83.82 | 77.94 | 71.47 | 68.53 | 75.44 | 73.96 | 68.54 | 69.35 | 71.57 | 70.62 |
> > | InternVL3-8B + MTCT + SARR | 85.00 | 81.76 | 72.94 | 69.41 | 77.28 | 74.76 | 70.85 | 70.62 | 72.11 | 72.09 |

---

> > > ### Author Response · Authors · 2025-11-22
> > >
> > > $\textbf{Response 2:}$ We first clarify that RD-MLDG is a **task-agnostic training framework**. Both MTCT and SARR are built upon structured reasoning-chain supervision rather than any task-specific architecture. As long as a task follows a ''vision → reasoning → output'' computational pathway, RD-MLDG can be directly applied without modifying the model architecture or the optimization objectives.
> > >
> > > In the main paper, we adopt the DomainBed setting because it is the most widely used and standardized benchmark for studying domain generalization. Building on this foundation allows us to investigate reasoning-chain supervision within a well-established DG framework, ensuring that our analysis of the challenges introduced by reasoning in DG is performed under consistent and community-recognized evaluation conditions. Through this setting, we identify and study two fundamental challenges caused by reasoning in DG: (i) the optimization difficulty and high-entropy gradients (Sec. 4.1), and (ii) the reasoning-pattern mismatch between external LLMs and target multimodal models (Sec. 4.2). MTCT and SARR are explicitly designed to address these two issues, and their mechanisms are independent of the specific downstream task used to study them.
> > >
> > > **Extension to VQA and VE:** To verify the task generality of RD-MLDG beyond visual classification, we extend our method to two multimodal tasks: VQA and Visual Entailment (VE). We construct reasoning-augmented versions of both VOLDOGER-VQA and VOLDOGER-VE \[1\] so that their data formats align with DomainBed-Reasoning. Each sample is augmented with the same five structured components: $\texttt{\<SUMMARY\>}$, $\texttt{\<CAPTION\>}$, $\texttt{\<REASONING\>}$, $\texttt{\<REFLECTION\>}$, and $\texttt{\<CONCLUSION\>}$. This ensures that both MTCT and SARR can be applied without architectural changes to any model.
> > > Both datasets contain four visually heterogeneous domains (Real, Cartoon, Pencil, Oil). The style shifts significantly affect visual perception, textual grounding, and multimodal alignment. This makes VQA and VE natural testbeds for assessing whether reasoning-driven DG generalizes beyond classification.
> > >
> > > **Experiment result:** Across both the VQA and VE tasks (see **Tab. R.1**), our results consistently demonstrate that RD-MLDG extends well beyond visual classification and remains effective in multimodal settings that require joint visual grounding and reasoning. On the VQA task, InternVL3-8B improves from 73.60\% (reasoning-only baseline) to 75.44\% with MTCT and further to 77.28\% with SARR. On the VE task, the same model improves from 70.59\% to 70.62\% and finally to 72.09\% after applying both modules. These gains are not isolated to a single model: LLaVA-1.5-7B, which lacks any reasoning-oriented pretraining, shows even larger improvements, rising from 63.68\% → 69.41\% → 71.62\% on VQA and from 67.12\% → 68.08\% → 70.13\% on VE. InternVL3-2B exhibits a similar trend on both tasks.
> > >
> > > Beyond absolute gains, RD-MLDG also enhances the competitiveness of the underlying models relative to stronger multimodal LLMs. After applying MTCT and SARR, InternVL3-8B surpasses most zero-shot MLLMs, including BLIP2, the Claude 3 family, Gemini 1.0 and 1.5, and several LLaVA-Next variants, in average accuracy across all four visual styles for both VQA and VE. The fact that these improvements appear consistently across two distinct multimodal tasks, each involving substantial style-induced domain shifts (Real, Cartoon, Pencil, Oil), further confirms that MTCT and SARR enable the model to learn domain-invariant reasoning patterns that remain stable across different cross-modal decision-making objectives.
> > >
> > > Taken together, the consistent performance gains across VQA and VE, coupled with improvements over a large set of open-source and proprietary multimodal baselines, provide strong evidence that RD-MLDG is a task-agnostic and broadly applicable reasoning-driven DG framework. The results confirm that our approach enhances the robustness and domain invariance of reasoning chains in diverse multimodal environments, thereby validating its generality far beyond visual classification.
> > >
> > > \[1\] *Choi, Juhwan, et al. "Voldoger: Llm-assisted datasets for domain generalization in vision-language tasks." Proceedings of the IEEE/CVF International Conference on Computer Vision. 2025.*

---

> > > > ### Author Response · Authors · 2025-11-22
> > > >
> > > > $\textbf{Weakness 3:}$ Baseline coverage: Comparisons are mainly against non-MLLM DG methods, making it difficult to assess the true advantage of RD-MLDG within the MLLM landscape.
> > > >
> > > > $\textbf{Response 3:}$ We appreciate the reviewer’s concern regarding baseline coverage. To clarify, RD‑MLDG is the first method to explicitly leverage MLLMs for domain generalization, and currently, no established DG baselines exist within the MLLM landscape.
> > > >
> > > > Despite this limitation, we ensured fair evaluation at the MLLM level. As shown in **Tab. 1 (main paper)**, RD-MLDG is compared against the strong commercial model GPT-4o and the open-source InternVL3-8B across all DomainBed datasets. RD‑MLDG consistently surpasses both models, demonstrating its effectiveness beyond traditional DG approaches.
> > > >
> > > > Furthermore, **Tab. 2 (main paper)** shows that our framework brings substantial improvements on both InternVL3-2B and InternVL3-8B under identical training and evaluation settings. These results demonstrate that RD-MLDG provides clear and fair gains within the MLLM landscape.
> > > >
> > > > $\textbf{Weakness 4:}$ (Minor) Writing: The introduction could benefit from citations to recent survey papers summarizing reasoning-based or MLLM-based DG approaches.
> > > >
> > > > $\textbf{Response 4:}$ We agree that survey papers can help contextualize a research direction. However, existing DG surveys do not include any reasoning-based or MLLM-based approaches, because these directions have not been explored in the DG literature. As stated in lines 77–78, our work is the first to introduce reasoning supervision and MLLMs into the DG setting.
> > > >
> > > > These two aspects should be regarded as components of a single new research direction, namely reasoning-enhanced domain generalization, which has not been covered by any prior survey. We will add a short remark in the introduction to make this context clear.
> > > >
> > > > $\textbf{Question 1 and Question 2:}$ Line 053: Please ensure the sentence ends with a period. Line 318: The description of how reasoning chains are combined with the classification prompt is unclear.
> > > >
> > > > $\textbf{Response 5:}$ We thank the reviewer for raising these points. 1) **Line 53:** We will carefully revise the final version to ensure the sentence ends with a period. 2) **Line 318:** In both the MTCT and SARR modules, the model receives two prompts: the reasoning prompt and the classification prompt. The key difference in the SARR module is that the reasoning prompt is replaced with a self-labeled reasoning chain, which has been refined during training. **For clarity, we have provided a detailed explanation of how the combination of these two prompts is implemented in the MTCT method, specifically between lines 295 and 299.** This section describes how both prompts are fed into the model and how the corresponding losses are jointly optimized.

---

> ### Comment · Reviewer_dvM6 · 2025-11-23
>
> Thank you for your detailed response, and most of the responses were satisfactory for me. Accordingly, I raised my rating to 4.
>
> Regarding the VOLDOGER paper, it seems like their motivation aligns with the author's, so it might be beneficial to discuss it in the related work section.
>
> ~~However, about the additional experiment with the VOLDOGER dataset, I could not find any released dataset about VOLDOGER-VQA and VOLDOGER-VE. I would like to ask where the authors acquired this dataset to perform the experiment.~~ Sorry for my mistake, I was able to find the dataset on GitHub.
>
> I am willing to further raise my score if the authors upload the revised paper with additional discussion and experiment.

---

> > ### Author Response · Authors · 2025-11-25
> >
> > $\textbf{Response:}$ Thank you for your detailed feedback. Regarding the VOLDOGER paper, we have added a discussion of this work in the related work section (see **Sec. 2**, from line 122 to line 125) of the revised manuscript, highlighting the similarities and differences between their approach and ours.
> >
> > We also appreciate your clarification regarding the VOLDOGER-VQA and VOLDOGER-VE datasets. In response to your comments, we have incorporated all the additional discussion and experiments from our response into the revised manuscript. This includes expanding the evaluation scope (see **Appendix E**), clarifying the methodological novelty (see **Sec. 5.2**, from line 313 to line 349), and improving baseline coverage.
> >
> > If there are any further questions or concerns, please do not hesitate to let us know. We are happy to address any remaining issues.

---

> > > ### Comment · Reviewer_dvM6 · 2025-11-25
> > >
> > > Thank you for your patient revision. After carefully reading it, I believe the paper has been more publishable. Accordingly, I updated my review.
> > >
> > > Some minor comments:
> > > - Line 213: Missing quotation mark " at the end of the sentence.
> > > - Table 7: "↑5.84%p" and "↑12.52%p" instead of just %.
> > > - Line 917: Wrong direction of quotation mark.

---

> > > > ### Author Response · Authors · 2025-11-25
> > > >
> > > > $\textbf{Response:}$ We greatly appreciate that you have raised your score and acknowledged the contribution of our work. Thank you for your thoughtful feedback and the time you've spent helping to refine our paper. In response to your suggestions, we have made the following revisions in the revised manuscript:
> > > > 1. **Line 213: Missing quotation mark " at the end of the sentence:** We have removed the extra quotation mark at the end of the sentence in line 213 (line 264 in the revised version).
> > > > 2. **Table 7: "↑5.84%p" and "↑12.52%p" instead of just %:** We have updated Tab 7 (**Appendix D**) to use ''↑5.84%p'' and ''↑12.52%p'' to correctly indicate percentage point changes, as you suggested. Additionally, we have made similar updates in Tab 2 (**main paper**) and Tab 5 (**Appendix B**), along with the corresponding analyses below, to ensure consistency across all tables and improve the clarity of the reported results.
> > > > 3. **Line 917: Wrong direction of quotation mark:** We have corrected the direction of the quotation mark at line 917 (line 917 in the revised version).

---

### Official Review · Reviewer_b6ME · 2025-10-29

**Soundness:** 3
**Presentation:** 3
**Contribution:** 3
**Rating:** 6
**Confidence:** 3

**Summary:**

This paper explores the integration of reasoning supervision into DG within MLLMs. The authors construct DomainBed-Reasoning, a new benchmark where each sample in DomainBed is paired with structured reasoning chains generated by GPT-4o. To address two key challenges -- (1) the optimization difficulty of reasoning-chain supervision and (2) the mismatch between external (GPT-4o) and self-generated reasoning patterns -- the authors introduce RD-MLDG. It comprises two components: Multi-Task Cross-Training (MTCT), which jointly optimizes classification and reasoning pathways to stabilize learning, and Self-Aligned Reasoning Regularization (SARR), a self-labeling mechanism that aligns supervision with the model’s own reasoning style. Experiments on four DG benchmarks (PACS, VLCS, OfficeHome, TerraIncognita) show consistent improvements over strong CLIP- and MLLM-based baselines. The paper argues that reasoning can serve as a complementary, process-level signal for domain generalization.

**Strengths:**

1. The work connects reasoning in MLLMs with robustness under domain shift, introducing a conceptually novel direction -- process-level invariance -- that goes beyond traditional feature-invariance approaches.

2. DomainBed-Reasoning is a non-trivial extension with structured reasoning chains, multi-stage generation, and rejection sampling to ensure coherence. This dataset can serve as a testbed for future studies on reasoning-based generalization.

3. RD-MLDG addresses two empirically observed issues with distinct and complementary mechanisms (MTCT for optimization stability, SARR for reasoning alignment). The framework is simple yet principled.

4. The authors provide extensive ablations, token-level entropy analyses, and convergence diagnostics that convincingly support the claimed effects of MTCT and SARR.

5. The paper is well-written, visually organized, and provides enough procedural detail for replication.

**Weaknesses:**

1. DomainBed-Reasoning relies entirely on GPT-4o-generated reasoning chains. These synthetic sequences likely encode stylistic and distributional priors from GPT-4o’s pretraining, rather than domain-grounded reasoning. As a result, RD-MLDG might learn to imitate linguistic style alignment rather than to capture transferable causal or process-level invariances. While the dataset is well-constructed, it remains uncertain whether the performance gains derive from genuine reasoning integration or from the regularizing effect of textual augmentation.

2. The paper treats reasoning chains as structured textual supervision but does not formally define what constitutes “reasoning” in the DG context -- e.g., whether it implies causal inference, compositional abstraction, or hierarchical explanation. Without such formal grounding or reasoning-quality diagnostics (e.g., logical coherence or factual correctness), it is difficult to evaluate whether RD-MLDG truly improves reasoning-driven generalization or simply leverages additional text-conditioned supervision.

3. The improvements attributed to reasoning may partly arise from multi-task regularization and self-training dynamics rather than reasoning-specific mechanisms. For example, MTCT’s stabilization could be viewed as a standard auxiliary-task regularization effect. SARR’s iterative self-labeling closely resembles confidence-based pseudo-labeling, widely used in semi-supervised and DG settings.
Demonstrating qualitative differences (e.g., changes in internal representation structure or reasoning coherence) would strengthen the claim that RD-MLDG explicitly enhances reasoning.

4. The method assumes that reasoning text provides process-level invariance, but the reasoning generation pipeline (via GPT-4o) is independent of the underlying visual feature encoder. This leads to potential semantic misalignment between visual features and textual reasoning tokens. While the model benefits empirically, the lack of explicit cross-modal alignment may limit scalability to other MLLM architectures or reasoning tasks.

5.  The four benchmarks (PACS, VLCS, OfficeHome, TerraIncognita) mainly test appearance-level domain shifts. Thus, the improvements reflect robustness to visual variation, not necessarily reasoning robustness. Experiments involving compositional or conceptual domain shifts (e.g., cross-task reasoning transfer or counterfactual settings) would better substantiate the central claim that reasoning contributes to domain generalization.

**Questions:**

See Weakness.

Additional Questions:
1. How do you evaluate the quality and logical validity of reasoning chains beyond coherence filtering?

2. Would RD-MLDG still outperform baselines if the reasoning text were semantically perturbed or replaced with neutral descriptions (to test for stylistic bias)?

3. Could MTCT or SARR generalize to smaller open-source models without reasoning pretraining (e.g., LLaVA-1.5) or purely visual transformers?

4. Do the authors observe changes in reasoning chain complexity or diversity across SARR iterations, and how does this correlate with accuracy gains?

---

> ### Author Response · Authors · 2025-11-22
>
> We sincerely thank the reviewer for the valuable questions. The Weaknesses \& Questions and Responses are as follows.
>
> $\textbf{Weakness 1 and Weakness 2 and Question 1 and Question 2:}$ The paper treats reasoning chains as structured textual supervision but does not formally define what constitutes “reasoning” in the DG context. How do you evaluate the quality and logical validity of reasoning chains beyond coherence filtering? Would RD-MLDG still outperform baselines if the reasoning text were semantically perturbed or replaced with neutral descriptions (to test for stylistic bias)?
>
> $\textbf{Response 1:}$ We thank the reviewer for raising this point, and we address the four related questions point by point as follows.
>
> **\(A\) On the definition of ''reasoning'':** In our work, we do not impose a formal taxonomy of reasoning types (e.g., causal, compositional, or hierarchical). Following multimodal reasoning frameworks such as LLaVA-CoT \[1\], we adopt an operational definition in which a reasoning chain is a structured, step-by-step textual explanation linking visual evidence to the final prediction. In practice, the generated chains naturally exhibit multiple forms of reasoning—causal (e.g., ''the glowing eyes indicate it is nocturnal''), compositional (e.g., combining pointed ears, slender appearance, and tail shape to narrow the label space), and hierarchical (e.g., coarse-to-fine identification such as ''belongs to the feline family → bobcat''). These forms emerge organically from the visual content rather than from explicit categorization.
>
> **\(B\) On reasoning-quality diagnostics:** To evaluate the quality and logical validity of reasoning chains beyond basic coherence filtering, we impose both structural and semantic constraints during dataset construction. A reasoning chain is retained only if it includes all five required components ($\texttt{\<SUMMARY\>}$, $\texttt{\<CAPTION\>}$, $\texttt{\<REASONING\>}$, $\texttt{\<REFLECTION\>}$, $\texttt{\<CONCLUSION\>}$) and its $\texttt{\<CONCLUSION\>}$ exactly matches the ground-truth label, which removes chains that appear structurally complete but are logically inconsistent. Building on the prompt design in LLaVA-CoT \[1\], we explicitly incorporate a $\texttt{\<REFLECTION\>}$ section to encourage GPT-4o to self-check its intermediate reasoning and improve the alignment between reasoning steps and the final answer. Furthermore, following Appendix B of LLaVA-CoT (''Prompt for data verification''), we verify that each retained $\texttt{\<CONCLUSION\>}$ is semantically supported by the preceding reasoning. Together, these semantic-level checks ensure that logical validity is enforced beyond mere structural coherence.
>
> **Table R.1: Comparison between standard reasoning supervision and neutralized reasoning supervision**
> | Model                                     | Location 100 | Location 38 | Location 43 | Location 46 |  Avg   |
> |-------------------------------------------|:------------:|:-----------:|:-----------:|:-----------:|:------:|
> | InternVL3-8B + CLS only                   |    72.80     |    65.64    |    69.70    |    58.62    |  66.69 |
> | InternVL3-8B + Reasoning only (Baseline)  |    74.42     |    60.39    |    68.21    |    55.21    |  64.56 |
> | InternVL3-8B + Reasoning only (Neutral)   |    70.39     |    57.13    |    64.44    |    52.27    |  61.06 |
>
> **\(C\) On distinguishing semantic reasoning from stylistic bias:** To directly evaluate whether RD-MLDG benefits from semantic reasoning rather than stylistic patterns, we performed an ablation where we replaced the original $\texttt{\<REASONING\>}$ content with a GPT-4o generated neutral description that preserves the surface form of reasoning but removes task-relevant semantics: $\texttt{\<REASONING\>}$\[I considered various visual aspects in the image, including general color patterns, texture variations, and broad shape cues. Although these observations do not clearly indicate any category, after synthesizing these ambiguous elements, I subjectively judge the image to belong to the XXX category.\]$\texttt{\</REASONING\>}$ while keeping the $\texttt{\<CONCLUSION\>}$ unchanged. As shown in the **Tab. R.1**, neutralizing the reasoning content leads to a consistent performance drop across TerraInc (Avg. 64.56\% → 61.06\%). This result demonstrates that RD-MLDG does not rely on generic text-conditioned supervision or GPT-4o's stylistic patterns; instead, it benefits from the semantic contribution of reasoning and the optimization mechanisms introduced by MTCT and SARR. Moreover, **Tab. 2 (main paper)** already shows that simply fine-tuning on raw reasoning text (SFT) does not produce comparable gains, further confirming that our improvements are reasoning-driven rather than style-driven.
>
> \[1\] *Xu, Guowei, et al. "Llava-cot: Let vision language models reason step-by-step." Proceedings of the IEEE/CVF International Conference on Computer Vision. 2025.*

---

> > ### Author Response · Authors · 2025-11-22
> >
> > $\textbf{Weakness 3 and Question 4:}$ The improvements attributed to reasoning may partly arise from multi-task regularization and self-training dynamics rather than reasoning-specific mechanisms. For example, MTCT's stabilization could be viewed as a standard auxiliary-task regularization effect. SARR's iterative self-labeling closely resembles confidence-based pseudo-labeling, widely used in semi-supervised and DG settings. Demonstrating qualitative differences (e.g., changes in internal representation structure or reasoning coherence) would strengthen the claim that RD-MLDG explicitly enhances reasoning. Do the authors observe changes in reasoning chain complexity or diversity across SARR iterations, and how does this correlate with accuracy gains?
> >
> > **Table R.2: Conclusion without image input (reasoning-only inference)**
> > | Model                         | Location 100 | Location 38 | Location 43 | Location 46 |  Avg.   |
> > |-------------------------------|:------------:|:-----------:|:-----------:|:-----------:|:------:|
> > | InternVL3-8B + Reasoning only |    71.11     |    55.24    |    65.91    |    53.14    |  61.35 |
> > | InternVL3-8B + MTCT           |    74.22     |    57.96    |    67.14    |    57.67    |  63.75 |
> > | InternVL3-8B + MTCT + SARR    |    79.16     |    64.45    |    72.14    |    61.01    |  69.19 |
> >
> > **Table R.3: Conclusion with image input (standard inference)**
> > | Model                         | Location 100 | Location 38 | Location 43 | Location 46 |  Avg.   |
> > |-------------------------------|:------------:|:-----------:|:-----------:|:-----------:|:------:|
> > | InternVL3-8B + Reasoning only |    74.42     |    60.39    |    68.21    |    55.21    |  64.56 |
> > | InternVL3-8B + MTCT           |    76.78     |    60.55    |    71.94    |    59.47    |  67.19 |
> > | InternVL3-8B + MTCT + SARR    |    81.74     |    66.12    |    73.38    |    61.36    |  70.65 |
> >
> > $\textbf{Response 2:}$ Thanks for the valuable comment. To verify that RD-MLDG improves reasoning rather than acting as generic regularization, we evaluate InternVL3-8B under two inference settings. In the Conclusion w/o image setting (see **Tab. R.2**), where the image is masked during the generation of the final $\texttt{\<CONCLUSION\>}$, accuracy improves from 61.35\% (Reasoning only) to 63.75\% with MTCT and to 69.19\% with MTCT+SARR. Because no visual information is provided, this substantial gain shows that the refined reasoning chains themselves become more discriminative and class-relevant after applying RD-MLDG.
> >
> > In the Conclusion w/ image setting (see **Tab. R.3 or Tab. 2 (main paper)**), we observe the same monotonic trend: accuracy increases from 64.56\% to 67.19\%, and further to 70.65\% with SARR. This parallel improvement in both masked and standard inference indicates that enhanced reasoning chains not only support prediction on their own, but also yield more stable and coherent multimodal inference when paired with visual evidence.
> >
> > Taken together, these results reveal a clear mechanism. MTCT and SARR progressively eliminate the reasoning-pattern mismatch between GPT-4o–style supervision and the model's own reasoning tendencies, steering the model toward more class-relevant tokens and away from stylistic or background-related text. Across SARR iterations, the reasoning chains shift from verbose and mismatched descriptions to concise and semantically discriminative explanations. This shift constitutes a qualitative change in the reasoning process itself rather than an arbitrary change in complexity or diversity, and it aligns precisely with the observed accuracy gains, especially the large improvement at N = 1. In addition, SARR is not a form of pseudo-labeling. It does not alter labels; instead, it restructures and refines the reasoning chain itself.
> >
> > Thus, the enhanced semantic quality and predictive power of the reasoning chains provide strong, direct evidence that RD-MLDG explicitly improves reasoning, rather than relying on auxiliary-task regularization or self-training dynamics.

---

> > > ### Author Response · Authors · 2025-11-22
> > >
> > > $\textbf{Weakness 4:}$ The method assumes that reasoning text provides process-level invariance, but the reasoning generation pipeline (via GPT-4o) is independent of the underlying visual feature encoder. This leads to potential semantic misalignment between visual features and textual reasoning tokens. While the model benefits empirically, the lack of explicit cross-modal alignment may limit scalability to other MLLM architectures or reasoning tasks.
> > >
> > > $\textbf{Response 3:}$ We thank the reviewer for the insightful observation. We would like to clarify that, during both MTCT and SARR, the multimodal model is fully fine-tuned: **the visual encoder and the language model are jointly updated**. Thus, the learning of reasoning chains is not decoupled from the visual representation space. While GPT-4o–generated reasoning is independent of the target model’s visual encoder, this semantic gap is precisely the challenge we analyze in Sec. 4.2. As shown there, GPT-4o reasoning exhibits a reasoning-pattern mismatch with InternVL3, and SARR is designed specifically to mitigate this issue. By progressively replacing GPT-4o reasoning with self-generated reasoning produced by the same visual–language architecture, SARR implicitly aligns reasoning tokens with the model’s own visual features, thereby reducing potential semantic misalignment. The consistent gains on both InternVL3-2B and InternVL3-8B in **Tab. 2 (main paper)** further demonstrate that this implicit alignment mechanism generalizes across MLLM backbones.

---

> > > > ### Author Response · Authors · 2025-11-22
> > > >
> > > > $\textbf{Weakness 5:}$ The four benchmarks (PACS, VLCS, OfficeHome, TerraIncognita) mainly test appearance-level domain shifts. Thus, the improvements reflect robustness to visual variation, not necessarily reasoning robustness. Experiments involving compositional or conceptual domain shifts (e.g., cross-task reasoning transfer or counterfactual settings) would better substantiate the central claim that reasoning contributes to domain generalization.
> > > >
> > > > **Table R.4: Base-to-new generalization results on FGVC-Aircraft**
> > > > | FGVC-Aircraft                         | Base  | New   | H     |
> > > > |--------------------------------------|:-----:|:-----:|:-----:|
> > > > | InternVL3-8B (Zero shot)              | 20.35 | 17.83 | 19.01 |
> > > > | InternVL3-8B + Reasoning only (Baseline) | 39.20 | 15.44 | 22.11 |
> > > > | InternVL3-8B + MTCT                   | 56.42 | 18.60 | 27.95 |
> > > > | InternVL3-8B + MTCT + SARR            | 60.11 | 24.33 | 34.63 |
> > > >
> > > > $\textbf{Response 4:}$ We thank the reviewer for the thoughtful observation. It is true that PACS, VLCS, OfficeHome, and TerraInc primarily capture appearance-level domain shifts, following the standard DG evaluation protocol. Our core objective, however, is to investigate whether class-relevant reasoning supervision can improve robustness under such realistic visual variations. To further examine whether reasoning provides benefits beyond appearance-level robustness, we additionally evaluate RD-MLDG under a conceptual generalization setting.
> > > >
> > > > To directly address the reviewer's concern, we conduct a base-to-new class generalization experiment on the FGVC-Aircraft dataset. Unlike the appearance-driven shifts in the four DomainBed benchmarks, the base-to-new protocol introduces a semantic and compositional shift in the label space: the model must transfer reasoning from seen categories to structurally distinct and previously unseen ones. FGVC-Aircraft \[1\] is suitable for this evaluation because its category distinctions arise from fine-grained structural and compositional attributes (e.g., wing geometry, engine configuration), making successful generalization rely more on structured reasoning than on superficial appearance cues.
> > > >
> > > > **Experiment setting:** Concretely, following existing work \[2\] that adopts the widely used base-to-new evaluation setting, we randomly split the 100 aircraft categories into 50 base and 50 new classes. All models are trained only on the base classes, and at test time we evaluate them on both base and unseen new classes, reporting performance on each split as well as their harmonic mean $H$. This setting ensures that good performance requires not only retaining accuracy on seen categories, but also transferring learned reasoning to semantically novel and compositionally distinct categories, thereby creating a genuine conceptual generalization scenario.
> > > >
> > > > **Observation:** As shown in the **Tab. R.4**, RD-MLDG improves harmonic mean accuracy from 19.01\% → 34.63\%, representing a substantial gain under this challenging conceptual shift. Notably, accuracy on unseen new classes increases from 15.44\% → 24.33\%, demonstrating that reasoning supervision enhances transfer to semantically novel and compositionally distinct categories rather than merely improving robustness to stylistic or appearance-level variations. These results provide direct empirical evidence that **RD-MLDG benefits conceptual generalization beyond the appearance-level shifts captured by existing benchmarks**. We regard more demanding compositional or counterfactual reasoning settings as promising directions for future work, and our framework offers a principled foundation for such extensions.
> > > >
> > > > \[1\] *Maji, Subhransu, et al. "Fine-grained visual classification of aircraft." arXiv preprint arXiv:1306.5151 (2013).*
> > > > \[2\] *Zhou, Kaiyang, et al. "Learning to prompt for vision-language models." International Journal of Computer Vision 130.9 (2022): 2337-2348.*

---

> > > > > ### Author Response · Authors · 2025-11-22
> > > > >
> > > > > $\textbf{Question 3:}$ Could MTCT or SARR generalize to smaller open-source models without reasoning pretraining (e.g., LLaVA-1.5) or purely visual transformers?
> > > > >
> > > > > **Table R.5: TerraInc results of MTCT and SARR**
> > > > > | Model                                   | Location100 | Location38  | Location43  | Location46  | Avg.  |
> > > > > |:----------------------------------------|:------:|:------:|:------:|:------:|:------:|
> > > > > | InternVL3-2B + Reasoning only (Baseline) | 74.79 | 61.26 | 69.40 | 58.26 | 66.00 |
> > > > > | InternVL3-2B + MTCT                      | 77.82 | 65.03 | 72.71 | 59.41 | 68.74 |
> > > > > | InternVL3-2B + MTCT + SARR               | 82.43 | 66.54 | 73.62 | 61.15 | 70.94 |
> > > > > | InternVL3-8B + Reasoning only            | 74.42 | 60.39 | 68.21 | 55.21 | 64.56 |
> > > > > | InternVL3-8B + MTCT                       | 76.78 | 60.55 | 71.94 | 59.47 | 67.19 |
> > > > > | InternVL3-8B + MTCT + SARR                | 81.74 | 66.12 | 73.38 | 61.36 | 70.65 |
> > > > > | LLaVA-1.5-7B + Reasoning only (Baseline)  | 75.34 | 58.20 | 62.51 | 52.24 | 62.07 |
> > > > > | LLaVA-1.5-7B + MTCT                        | 79.87 | 58.93 | 63.38 | 53.48 | 63.92 |
> > > > > | LLaVA-1.5-7B + MTCT + SARR                 | 80.93 | 60.01 | 64.51 | 55.14 | 65.15 |
> > > > >
> > > > > $\textbf{Response 5:}$ Thanks for the valuable comment. To evaluate whether MTCT and SARR can generalize to smaller open-source models without dedicated reasoning pretraining, we conducted additional experiments on LLaVA-1.5-7B, whose reasoning ability is substantially weaker than that of InternVL3. This is directly reflected in the reasoning-only baselines: under the same setting on TerraIncognita, InternVL3-2B achieves an average accuracy of 66.00\%, whereas LLaVA-1.5-7B reaches only 62.07\%, indicating a clear gap in their initial reasoning capability. Despite this weaker backbone, **our method remains consistently effective**. As shown in the **Tab. R.5**, MTCT improves LLaVA-1.5-7B from 62.07\% to 63.92\%, and adding SARR further increases performance to 65.15\%. A similar improvement is observed on InternVL3-2B, indicating that both components provide stable gains even when the underlying model lacks strong reasoning pretraining.
> > > > >
> > > > > These results collectively demonstrate that RD-MLDG does not rely on a strong reasoning backbone. **Both MTCT and SARR transfer effectively to smaller, weaker, and fully open-source multimodal models**, showing that the proposed framework is broadly applicable beyond large reasoning-centric models.

---

### Official Review · Reviewer_Zr5a · 2025-11-01

**Soundness:** 3
**Presentation:** 3
**Contribution:** 3
**Rating:** 6
**Confidence:** 3

**Summary:**

This paper tackles the problem of domain generalization (DG) by proposing a novel approach that leverages the reasoning capabilities of Multimodal Large Language Models (MLLMs). Instead of relying on traditional methods that seek feature-level invariance, the authors argue for pursuing process-level invariance by training models to generate class-relevant reasoning chains, which are hypothesized to be more robust to domain shifts. The paper proposes RD-MLDG that jointly optimizes a direct classification path and the reasoning-generation path, and introduces an iterative self-labeling stage where the model generates its own reasoning chains, which are then filtered and used as the new supervision signal. The authors demonstrate that RD-MLDG achieves state-of-the-art results on four standard DG benchmarks (PACS, VLCS, OfficeHome, and TerraIncognita).

**Strengths:**

1. The paper addresses domain generalization by proposing a novel approach that leverages the reasoning capabilities of Multimodal Large Language Models, which is a challenging and practical scenario.
2. The paper is well written and easy to follow.
3. The paper provides extensive experiments, showing the effectiveness and versatility of the proposed method.

**Weaknesses:**

1. The quality of the entire DomainBed-Reasoning dataset influences on reasoning chains generated by GPT-4o. This introduces a potential dependency and bias.

2. The proposed training procedure appears computationally intensive. It involves an initial MTCT stage followed by N=3 rounds of SARR. Each SARR round seems to require a full generation pass over the source data, a filtering step, and another fine-tuning stage.

**Questions:**

1. In the SARR filtering step, what percentage of self-generated reasoning chains are typically discarded for leading to an incorrect conclusion? Does this rejection rate decrease as the SARR rounds progress?

2. The core hypothesis is that reasoning chains are more domain-invariant than visual features. Figure 1 provides a great qualitative example. Have you considered any quantitative validation of this hypothesis? For instance, one could measure the embedding-space similarity of reasoning chains for the same class across different domains, and compare this to the similarity of visual features for that same class across domains.

---

> ### Author Response · Authors · 2025-11-22
>
> We sincerely thank the reviewer for the valuable questions. The Weaknesses \& Questions and Responses are as follows.
>
> $\textbf{Weakness 1:}$ The quality of the entire DomainBed-Reasoning dataset influences the reasoning chains generated by GPT-4o, introducing potential dependency and bias.
>
> $\textbf{Response 1:}$ We appreciate the reviewer's observation. We agree that GPT-4o reasoning chains inevitably carry stylistic priors that may introduce dependency and bias. Importantly, we explicitly analyze this issue in **Sec. 4.2 (“Mismatches in Reasoning Patterns Across Sources”)**, where we quantify the mismatch between GPT-4o–generated supervision and the model’s own reasoning distribution.
>
> Our analysis shows that GPT-4o reasoning is rich but context-heavy, whereas InternVL3-8B naturally produces shorter, label-focused chains. When fine-tuning with GPT-4o reasoning, only +1.88% of tokens exceed a confidence threshold of 0.75; in contrast, self-generated reasoning yields a +29.74% increase (**Fig. 4.A (main paper)**). Moreover, the top-15 tokens with the largest entropy reduction (**Fig. 4.B–C (main paper)**) reveal a clear divergence in semantic focus: GPT-4o emphasizes descriptive background attributes, while the model itself emphasizes category-discriminative cues. This confirms that GPT-4o supervision introduces a distributional mismatch rather than random noise.
>
> **SARR is designed to mitigate this dependency.** By iteratively retaining only self-labeled reasoning chains with correct conclusions, SARR progressively replaces GPT-4o–style stylistic tokens with model-aligned, class-relevant reasoning while preserving useful semantics. The supervision becomes increasingly consistent with the model’s internal reasoning tendencies, reducing reliance on GPT-4o priors.
>
> This mitigation is also reflected in downstream performance. As shown in **Tab. 2 (main paper)**, moving from GPT-4o reasoning to self-aligned reasoning consistently improves generalization: for InternVL3-2B, +0.65% on OfficeHome and +2.20% on TerraInc; for InternVL3-8B, +1.15% and +3.46%, respectively ($5^{th}$ row vs. $7^{th}$ row and $11^{th}$ row vs. $13^{th}$ row). These gains stem from modifying only the reasoning supervision, confirming that SARR effectively reduces GPT-4o dependency and improves both the quality and optimizability of reasoning chains.

---

> > ### Author Response · Authors · 2025-11-22
> >
> > $\textbf{Weakness 2:}$ The proposed training procedure appears computationally intensive. It involves an initial MTCT stage followed by N = 3 rounds of SARR. Each SARR round seems to require a full generation pass over the source data, a filtering step, and another fine-tuning stage.
> >
> > **Table R.1: Performance across SARR iterations (N = 0 to 3) on the four TerraInc domains**
> > | N | Location 100 | Location 38 | Location 43 | Location 46 | Avg. |
> > |:-|:------------:|:-----------:|:-----------:|:-----------:|:----:|
> > | 0 | 76.78        | 60.55       | 71.94       | 59.47       | 67.19 |
> > | 1 | 81.06        | 65.56       | 72.81       | 60.80       | 70.06 |
> > | 2 | 81.62        | 65.93       | 73.49       | 61.34       | 70.59 |
> > | 3 | 81.74        | 66.12       | 73.38       | 61.36       | 70.65 |
> >
> > $\textbf{Response 2:}$ Thanks for the valuable comment. Our method uses relatively small VLM backbones, so the generation pass in each SARR round is negligible compared with the fine-tuning cost. We further provide quantitative evidence showing that even with fewer SARR rounds (smaller N) or reduced epochs per round, the approach continues to deliver strong gains.
> >
> > **Impact of fewer SARR rounds.** As shown in **Tab. R.1**, the first SARR round (N = 1) already provides a clear improvement (+2.87%) over N = 0, because it replaces GPT-4o reasoning with InternVL3-generated reasoning that matches the model’s own reasoning style (see **Fig. 7 (main paper)**). Additional rounds (N > 1) mainly increase the number of correctly self-labeled reasoning chains to expand model-consistent supervision, rather than repeatedly refining the model.
> >
> > **Table R.2: Ablation on SARR efficiency: varying the number of iterations (N) and epochs per iteration**
> > | Setting        | Location 100 | Location 38 | Location 43 | Location 46 | Avg. |
> > |:--------------|:------------:|:-----------:|:-----------:|:-----------:|:----:|
> > | N=1, epoch=3   | 81.06        | 65.56       | 72.81       | 60.80       | 70.06 |
> > | N=3, epoch=1   | 81.50        | 65.71       | 73.00       | 61.32       | 70.39 |
> > | N=3, epoch=3   | 81.74        | 66.12       | 73.38       | 61.36       | 70.65 |
> >
> > **Reducing epochs per round.** To further reduce cost, we evaluate N = 3 with only one epoch per SARR round. This configuration yields almost the same computational cost as training only N = 1 round. The performance (see **Tab. R.2**) remains close to the full three-epoch setting because MTCT must fit GPT-4o reasoning (which contains the reasoning-pattern mismatch), whereas after the first SARR round the supervision switches to InternVL3's own reasoning. Once this alignment happens, optimization becomes much easier and each round converges within a single epoch.
> >
> > $\textbf{Question 1:}$ In the SARR filtering step, what percentage of self-generated reasoning chains are discarded because they lead to an incorrect conclusion? Does this rejection rate decrease as SARR rounds progress?
> >
> > **Table R.3: Rejection rate across SARR iterations (N) on the four TerraInc domains**
> > | N | Location 100 | Location 38 | Location 43 | Location 46 |  Avg.   |
> > |:-|:------------:|:-----------:|:-----------:|:-----------:|:------:|
> > | 0 |    41.78     |    39.25    |    40.84    |    36.18    | 39.51  |
> > | 1 |    21.66     |    22.14    |    17.56    |    15.57    | 19.23  |
> > | 2 |    18.49     |    17.98    |    15.07    |    13.64    | 16.30  |
> > | 3 |    17.07     |    15.74    |    14.43    |    12.01    | 14.81  |
> >
> > $\textbf{Response 3:}$ Thanks for the valuable comment. To quantify the filtering behavior of SARR, we measure the rejection rate, defined as the percentage of self-generated reasoning chains whose $\texttt{\<CONCLUSION\>}$ does not match the ground-truth label. Using InternVL3-8B on TerraInc (see **Tab. R.3**), we observe a relatively high rejection rate at N = 0 (39.51%), reflecting the mismatch between GPT-4o–style reasoning and the model's native prediction tendencies. After the first SARR iteration, the rejection rate drops sharply to 19.23%, and continues decreasing at N = 2 and N = 3, stabilizing around 15–17%.
> >
> > This consistent downward trend shows that as SARR progresses, the model's self-generated reasoning chains increasingly yield correct conclusions. Instead of performing random filtering, SARR progressively strengthens the semantic alignment between the reasoning chain and the model’s own prediction. The improving correctness of self-generated chains indicates that the reasoning process itself becomes more stable, more coherent, and more predictive. This provides direct evidence that RD-MLDG enhances reasoning quality rather than relying on incidental regularization.

---

> > > ### Author Response · Authors · 2025-11-22
> > >
> > > $\textbf{Question 2:}$ The core hypothesis is that reasoning chains are more domain-invariant than visual features. Figure 1 provides a strong qualitative example. Have you considered any quantitative validation of this hypothesis? For instance, one could measure the embedding-space similarity of reasoning chains for the same class across different domains, and compare this with the similarity of visual features for the same class.
> > >
> > > **Table R.4: Cross-domain divergence of visual embeddings and reasoning-chain embeddings for each TerraInc class**
> > > | Class | bird | bobcat | cat | coyote | dog | empty | opossum | rabbit | raccoon | squirrel | Avg |
> > > |-----|:----:|:------:|:----:|:------:|:---:|:-----:|:--------:|:------:|:--------:|:--------:|:---:|
> > > | Domain divergence (Visual) | 0.209 | 0.251 | 0.314 | 0.387 | 0.213 | 0.144 | 0.266 | 0.167 | 0.258 | 0.176 | 0.239 |
> > > | Domain divergence (Text)   | 0.054 | 0.048 | 0.103 | 0.114 | 0.093 | 0.126 | 0.091 | 0.103 | 0.142 | 0.115 | 0.099 |
> > >
> > > $\textbf{Response 4:}$ Thanks for the valuable comment. Yes, we performed a quantitative validation of the hypothesis that reasoning chains are more domain-invariant than visual features. Following existing work that measures cross-domain distributional discrepancies \[1\], we compute the Maximum Mean Discrepancy (MMD) between a source domain $\hat D^S_m$ and a target domain $\hat D^T_{m'}$. Let $f(x)$ denote the embedding extracted from CLIP's vision encoder for visual features, or from CLIP’s text encoder for reasoning-chain embeddings. Under a linear kernel, MMD reduces to the squared distance between mean embeddings:
> > >
> > > $$
> > > \mathrm{MMD}^2(\hat D^S_m,\hat D^T_{m'})=||\frac{1}{N^s_m}\sum_{i=1}^{N^s_m} f(x_i)-\frac{1}{N^t_{m'}}\sum_{j=1}^{N^t_{m'}} f(x_j)||_2^2
> > > $$
> > >
> > > Using identical domain splits for both modalities, we compute the MMD for each category in TerraInc. As summarized in the **Tab. R.4**, reasoning-chain embeddings exhibit dramatically lower cross-domain divergence than visual embeddings (average 0.239 → 0.099, a 58.6\% reduction). This large gap provides clear quantitative evidence that reasoning chains capture semantically stable, class-relevant information that is far less sensitive to style, background, or environmental shifts than visual features. Therefore, the results strongly support our core hypothesis that reasoning chains are significantly more domain-invariant.
> > >
> > > \[1\] *Guo, Jintao, et al. "Aloft: A lightweight mlp-like architecture with dynamic low-frequency transform for domain generalization." Proceedings of the IEEE/CVF conference on computer vision and pattern recognition. 2023.*

---

> ### Comment · Reviewer_Zr5a · 2025-11-25
>
> Thanks the reviewer for the detailed response.  Most of my concerns are addressed and therefore I will maintain the positive score.

---

> > ### Author Response · Authors · 2025-11-26
> >
> > We sincerely thank the reviewer for the encouraging feedback and for maintaining the positive scores. We greatly appreciate your time and constructive comments.

---

### Author Response · Authors · 2025-12-03

We thank all reviewers for their careful evaluation and constructive feedback. **All reviewers acknowledged the novelty and significance of our work**:
- **Novel idea**: leveraging reasoning for domain generalization and introducing the concept of process-level invariance (**highlighted by Reviewers `Zr5a`, `b6ME`, and `jvkU`**);
- **Strong results**: the RD-MLDG framework and the DomainBed-Reasoning benchmark show clear and consistent improvements (**highlighted by all four reviewers: `Zr5a`, `b6ME`, `dvM6`, and `jvkU`**).

Importantly, Reviewer **`dvM6`** initially gave a negative score of 2, largely due to two major misunderstandings of our method. During the discussion period, they commented that **`''our rebuttal and revisions had resolved these concerns and made the paper more publishable''`**, and accordingly raised their score from 2 to 8, while Reviewer **`Zr5a`** confirmed that their main concerns were satisfactorily addressed and maintained a positive final assessment. Before the rollback to the initial scores, the ratings were **8/6/6/6 (average 6.5)**, reflecting that our responses effectively addressed the main concerns and that the contribution is viewed favorably overall.

___

### Declaration about the OpenReview bug

We did not use, propagate, or benefit from the OpenReview bug that exposed the identities of authors, reviewers, and area chairs.

Our last interactions with the four reviewers ended at the following times:

* Reviewer **`Zr5a`**: **`26 Nov 2025, 14:12`**
    * Initial score: 6 $\rightarrow$ Final score: 6
* Reviewer **`b6ME`**: **`23 Nov 2025, 00:05`**
* Reviewer **`dvM6`**: **`25 Nov 2025, 18:06`**
    * Initial score: 2 $\rightarrow$ Final score: 8
* Reviewer **`jvkU`**: **`23 Nov 2025, 00:14`**

---

### Reviewer-wise questions, our responses, and final reviewer comments

After our detailed rebuttal, Reviewer **`dvM6`** raised their scores, and Reviewer **`Zr5a`** kept a positive score. Below we summarize, for each reviewer, the main concerns we addressed and the final comments.

#### Reviewer **`dvM6`**

**Initial misunderstandings.** Reviewer **`dvM6's`** initial review, which assigned a score of 2, was largely driven by two misunderstandings of our method: (i) treating MTCT and SARR as essentially the same loss rather than distinct inner and outer components; and (ii) characterizing the MLLM-level baselines as limited. We corrected these misunderstandings by: (i) clarifying that **MTCT is an inner training objective, whereas SARR is an outer, multi-round self-alignment procedure** that reshapes the supervision distribution; and (ii) pointing out **existing MLLM-level comparisons in Tab. 2 (main paper)**, where RD-MLDG consistently improves over InternVL3-2B and InternVL3-8B under shared settings, and revising the text to make these results more prominent.

**Main concern.** Reviewer **`dvM6`** also raised a substantive concern that the evaluation scope is too narrow, since the experiments focus mainly on visual classification with relatively few multimodal tasks, making the method's role in the broader MLLM ecosystem less clear; in response, we broadened the evaluation by extending RD-MLDG to VOLDOGER-VQA and VOLDOGER-VE and observed consistent OOD gains across models (**Tabs. R.1–R.2 in the rebuttal**), which are now reflected in the revised method and experiments sections.

**Post-rebuttal comments from `dvM6`:**
The reviewer's post-rebuttal comments indicate that our responses resolved the main concerns and clarified earlier misunderstandings about our method, leading them to substantially raise their score and regard the paper as more publishable, as reflected below:

- *First post-rebuttal comment (23 Nov 2025, 21:54; **after our detailed response**)*
> Thank you for your detailed response, and most of the responses were satisfactory for me.

- *Second post-rebuttal comment (25 Nov 2025, 15:40; **after we uploaded the revised paper with additional discussion and experiments**)*
> Thank you for your patient revision. After carefully reading it, I believe the paper is more publishable. Accordingly, I updated my review.

---

> ### Author Response · Authors · 2025-12-03
>
> #### Reviewer **`Zr5a`**
>
> **Main concerns.** Reviewer **`Zr5a`** raised two related concerns: (i) that DomainBed-Reasoning relies entirely on GPT-4o–generated reasoning chains, potentially creating supervision dependency and bias; and (ii) that the MTCT + multi-round SARR pipeline may be computationally expensive.
>
> **Our responses.** In our rebuttal and revised version, we addressed these concerns as follows: (i) we analyzed the distributional mismatch between GPT-4o supervision and the model's own reasoning and showed that SARR progressively replaces GPT-4o–style supervision with self-generated, class-discriminative reasoning chains, reducing this dependency and improving downstream performance (**Tab. 2 in the main paper**); and (ii) we quantified the computational cost and showed that most gains come from the first SARR round, with a three-round, one-epoch configuration achieving performance close to the full setting at a cost comparable to a single-round variant (**Tabs. R.1–R.2 in the rebuttal**). We additionally provide quantitative analyses of domain invariance and SARR's filtering behavior across iterations (**Tabs. R.3–R.4**), which are now incorporated into the revised manuscript.
>
> **Post-rebuttal comments from `Zr5a`:**
> The reviewer confirmed that our rebuttal satisfactorily addressed their main concerns and explicitly decided to keep a positive score:
>
> - *First post-rebuttal comment (26 Nov 2025, 14:12; **after our detailed response**)*
> > Thanks for the detailed response. Most of my concerns are addressed and therefore I will maintain the positive score.
>
> ---
>
> #### Reviewer **`b6ME`**
>
> **Main concerns.** Reviewer **`b6ME`** raised three related concerns: (i) that our notion of ''reasoning'' in the DG setting was not sufficiently clear, and that the gains of RD-MLDG might mainly come from generic text-conditioned supervision, stylistic patterns, or multi-task / pseudo-labeling–like dynamics rather than genuine reasoning; (ii) that experiments focusing mainly on appearance-level shifts may not directly demonstrate robustness of reasoning; and (iii) that it was unclear whether the method transfers to smaller open-source models.
>
> **Our responses.** In our rebuttal and revised version, we addressed these concerns as follows: (i) we provided an explicit operational definition of reasoning and, via a neutralized-reasoning ablation and an SFT baseline (**Tab. R.1 in the rebuttal; Tab. 2 in the main paper**), showed that removing task-relevant reasoning content leads to clear performance drops, indicating that RD-MLDG's gains come from semantically meaningful reasoning supervision rather than surface style or generic text conditioning; (ii) we evaluated InternVL3-8B under both reasoning-only and standard multimodal inference and observed monotonic improvements from the ''Reasoning only'' baseline to MTCT and MTCT+SARR (**Tabs. R.2–R.3**), supporting robustness beyond appearance-level shifts and simple multi-task / pseudo-labeling dynamics; and (iii) we added a base-to-new class generalization study on FGVC-Aircraft and experiments on LLaVA-1.5-7B (**Tabs. R.4–R.5**), which show consistent gains on unseen classes and on a smaller open-source model, with these analyses incorporated into the revised manuscript.
>
> ---
>
> #### Reviewer **`jvkU`**
>
> **Main concerns.** Reviewer **`jvkU`** raised two related concerns: (i) that the description of the reasoning-chain generation and filtering process in DomainBed-Reasoning was not sufficiently concrete; and (ii) that we lacked evidence that RD-MLDG remains effective and efficient beyond image classification.
>
> **Our responses.** In our rebuttal and revised version, we addressed (i) by giving a concrete specification of the reasoning-chain pipeline (number of GPT-4o candidates per example, selection criteria, empirical rejection rate, and the role of the $\texttt{\<REFLECTION\>}$ token) in the revised manuscript, and (ii) by extending RD-MLDG to VQA and Visual Entailment on reasoning-augmented VOLDOGER-VQA and VOLDOGER-VE and adding a SARR-round/epoch ablation, showing that the method remains effective and computationally efficient beyond image classification (**Tabs. R.1–R.2 in the rebuttal**).

---

### Meta-Review · Area_Chair_vuhU · 2026-01-06

**Summary:**

This paper tackles the problem of domain generalization by proposing a novel approach that leverages the reasoning capabilities of multimodal large language models.  It is a non-trivial extension with structured reasoning chains, multi-stage generation, and rejection sampling to ensure coherence.The framework is simple yet principled. The authors provided extensive ablations, token-level entropy analyses, and convergence diagnostics that convincingly support the claimed effects of MTCT and SARR. The paper is well-written, visually organized, and provides enough procedural detail for replication.  During the discussion phase, the reviewer with rating 2 would like to increase it to 6--all reviewers appreciate the contribution of the paper and recommended acceptance.

**Reviewer Scores:**

No

---

### Decision · Program_Chairs · 2026-01-26

Accept (Poster)